# Active Reward Modeling: Adaptive Preference Labeling for Large Language Model Alignment

**Yunyi Shen** [* 1]   **Hao Sun** [* 2]   **Jean-François Ton** [3]

## Abstract

Building neural reward models from human preferences is a pivotal component in reinforcement learning from human feedback (RLHF) and large language model alignment research. Given the scarcity and high cost of human annotation, how to select the most informative pairs to annotate is an essential yet challenging open problem. In this work, we highlight the insight that an ideal comparison dataset for reward modeling should balance *exploration of the representation space* and make *informative comparisons* between pairs with moderate reward differences. Technically, challenges arise in quantifying the two objectives and efficiently prioritizing the comparisons to be annotated. To address this, we propose the Fisher information-based selection strategies, adapt theories from the *classical experimental design* literature, and apply them to the final linear layer of the deep neural network-based reward modeling tasks. Empirically, our method demonstrates remarkable performance, high computational efficiency, and stability compared to other selection methods from deep learning and classical statistical literature across multiple open-source LLMs and datasets. Further ablation studies reveal that incorporating cross-prompt comparisons in active reward modeling significantly enhances labeling efficiency, shedding light on the potential for improved annotation strategies in RLHF. Code and embeddings to reproduce all results of this paper are available at https://github.com/YunyiShen/ARM-FI/.

## 1. Introduction

The safe and successful deployment of Large Language Models (LLMs) across various application domains requires alignment with human values. Current research working on LLM alignment mainly focuses on reinforcement learning from human feedback (RLHF) (Christiano et al., 2017; Stiennon et al., 2020; Ouyang et al., 2022; Bai et al., 2022a), which rely on preference-based annotations provided by human annotators (Bai et al., 2022b). However, obtaining human feedback can be expensive, and the noisy and binary nature of such data often limits its information density, posing a challenge for effective reward modeling (Wang et al., 2024a; Liu et al., 2024).

Active learning, where the model queries most informative labels based on its current state, offers a potential solution. It typically involves three key components: an initial model, a query strategy — often in the form of maximizing a scoring function over unlabeled data, and a pool of unlabeled data. The model selects a subset of data for labeling and retrains iteratively until a stopping criterion is met.

In this work, we study the problem of active data acquisition in reward modeling. Technically, we introduce various scoring rules inspired by both *classical experimental design* (Chaloner & Verdinelli, 1995) and recent deep learning-based advancements (Sener & Savarese, 2017; Houlsby et al., 2011; Kirsch et al., 2019). We adapt those methods to the learning of Bradley-Terry (BT) reward models (Bradley & Terry, 1952), which have been successfully applied in large-scale alignment practices (Ouyang et al., 2022; Touvron et al., 2023) and proven to be theoretically sound (Sun et al., 2025a).

We benchmark 8 scoring algorithms using 2 datasets and 3 LLMs, ranging in size from 2B to 8B, and evaluate a wide range of active learning setups. Our results show that two classical experimental design methods — applied to the final linear feature layer of deep models — achieve state-of-the-art performance and strong stability across different setups, model architectures, and datasets.

Our main contributions can be summarized as follows:

- Formally, we characterize the problem of optimal prefer-

---

*Equal contribution  [1]EECS, MIT, Cambrdige, MA, USA [2]University of Cambridge, Cambrdige, UK [3]ByteDance Research, London, UK. Correspondence to: Yunyi Shen <yshen99@mit.edu>, Hao Sun <hs789@cam.ac.uk>, Jean-François Ton <jeanfrancois@bytedance.com>.

*Proceedings of the 42$^{nd}$ International Conference on Machine Learning*, Vancouver, Canada. PMLR 267, 2025. Copyright 2025 by the author(s).

ence label annotation using embedding space BT regression framework and establish connections between active learning and classical experimental design literature under the BT context.

- Methodologically, we introduce a set of algorithms inspired by classical experimental design literature, adapt them for deep BT regression models, and develop an efficient gradient approximation for the associated combinatorial optimization challenge in large-scale alignment problems.
- Empirically, we evaluate different methods for preference label annotation across diverse setups, datasets, and base models. Our results suggest that applying classical experimental design techniques to the final layer of a deep neural network yields strong performance and stability.

## 2. Background and Setup

**Reward modeling in alignment.** Reinforcement learning is a key technique for aligning LLMs to ensure their safe and effective deployment (Christiano et al., 2017; Ouyang et al., 2022; Stiennon et al., 2020). The most prevailing approach, RLHF, relies on reward models as a fundamental mechanism for quantifying content quality and scaling the reinforcement learning (Lambert et al., 2024; Wang et al., 2024a). During fine-tuning and deployment, reward models serve as proxies for human evaluators (Dubey et al., 2024; Dong et al., 2024; Wang et al., 2024b), assessing how well LLM outputs align with human intent. Despite significant progress, reward modeling remains challenging due to the scarcity and inaccuracy of annotations (Lambert et al., 2024; Wang et al., 2024a; Gao et al., 2023). Prior research has attempted to mitigate these challenges through different aspects when learning from a fixed set of annotations (Wang et al., 2024b; Winata et al., 2024; Liu et al., 2024; Lou et al., 2024; Coste et al., 2023; Zhang et al., 2024). While Xiong et al. (2023); Dong et al. (2024) demonstrate that online annotations are more efficient in RLHF, the topic of online annotation prioritization strategy remain underexplored except for heuristic designs (Muldrew et al., 2024).

**Bradley-Terry model for reward modeling.** A canonical model used for reward modeling from binary preference data is the Bradley-Terry (BT) model (Bradley & Terry, 1952), or more precisely, its regression variant (Sun et al., 2025a). In the most general setting, which allows for cross-prompt comparisons, a human annotator is presented with two pairs of prompts and responses, $(x_{i,1}, y_{i,1})$ and $(x_{i,2}, y_{i,2})$. The annotator then provides a preference, $h_i = 1_{\{(x_{i,1}, y_{i,1}) \succ (x_{i,2}, y_{i,2})\}}$ indicating whether the first pair is preferred over the second.

Often, both responses correspond to the same prompt, i.e., $x_{i,1} = x_{i,2}$ however, Bradley-Terry regression can operate without this assumption. The model regresses these anno-

tations onto an embedding $\Psi(x_{i,1}, y_{i,1})$. This is a mild assumption since these embeddings can be, for example, a concatenation of word embeddings, the output of tokenizers, or the output embedding of an LLM.

When there is no risk of confusion, we denote the embeddings of pair $i$ as $\Psi_{i,1} \in \mathbb{R}^D$ and $\Psi_{i,2} \in \mathbb{R}^D$, with a reward function $r : \mathbb{R}^D \to \mathbb{R}$. The goal is to learn this function from annotations. In the BT model, we assume that

$$h_i \sim \text{Bernoulli}\left(\sigma[r(\Psi_{i,1}) - r(\Psi_{i,2})]\right) \quad (1)$$

with $\sigma$ being the sigmoid function.

**Active learning.** In a typical active learning setting, we have a labeled dataset of size, $\mathcal{D}_s = (x_{i,1}, y_{i,1}, x_{i,2}, y_{i,2}, h_i)_{i=1}^{I_s}$ with sample size $I_s$ at step $s$, and a typically large pool of unlabeled data, $\mathcal{P}_s = (x_{j,1}, y_{j,1}, x_{j,2}, y_{j,2})_j^{J_s}$ with sample size $J_s$ at step $s$. The goal is to select a small subset $\mathcal{C}_s \subset \mathcal{P}_s$, subject to certain constraints, for labeling. Once labeled (denoted as $\tilde{\mathcal{C}}_s$), this subset is added to the labeled dataset to train the next iteration of the model.

We also consider this process in the embedding space, where the labeled and unlabeled sets are given by $\mathcal{D}_s = \{(\Psi_{i,1}, \Psi_{i,2}, h_i)\}_{i=1}^{I_s}$ and $\mathcal{P}_s = \{(\Psi_{j,1}, \Psi_{j,2})\}_{j=1}^{J_s}$. A typical model-based active learning procedure is outlined in Algorithm 1. In this work, we focus on identifying the best-performing scoring rules.

---

**Algorithm 1** Model-based active learning

---

**Require:** initial labeled dataset $\mathcal{D}_0$, pool set $\mathcal{P}_0$, model $\mathcal{M}_0$, a scoring rule $\mathcal{S}$, budget constrain $c$, and number of rounds $n$
1: **RETURN** Last trained model $\mathcal{M}_n$
2: **for** $s \leftarrow 1$ to $n$ **do**
3:      generate pool $\mathcal{P}_s$
4:      $\mathcal{C}_s \leftarrow \text{argmax}_{\mathcal{C} \subset \mathcal{P}_s, |\mathcal{C}| \leq c} \mathcal{S}(\mathcal{M}_{s-1}, \mathcal{C}, \mathcal{D}_{s-1})$
5:      get labeled dataset $\tilde{\mathcal{C}}_s$
6:      $\mathcal{D}_s \leftarrow \tilde{\mathcal{C}}_s \cup \mathcal{D}_{s-1}$
7:      train model $\mathcal{M}_s$ using $\mathcal{D}_s$
8: **end for**
9: **return** $\mathcal{M}_n$

---

In literature, Muldrew et al. (2024) considered active learning and proposed a strategy that combines entropy with model certainty (which is equivalent to the maxdiff strategy in our notation). For non-binary data, Mukherjee et al. (2024) suggested maximizing the determinant of the feature matrix. BatchBALD (Kirsch et al., 2019) is a general-purpose active learning algorithm that requires a Bayesian model. The scoring in this method aims to maximize the expected entropy reduction by selecting the most informative data points. Experimental design for generalized linear models has been extensively studied in the classical statistical literature, with logistic regression serving as a key

example (see e.g., Chaloner & Verdinelli, 1995; Sener & Savarese, 2017). Under the assumption of a linear reward function, the Bradley-Terry (BT) model simplifies to logistic regression. We acknowledge a concurrent work Feng et al. (2025) also used Fisher information (FI) as one of their tools in sample selection for DPO. Feng et al. (2025) used FI of all parameters in their neural net to establish theory on their sampling based algorithm but did not directly as optimization target in sample selection. While this paper compared many different strategies empirically and suggests determinant of FI of last layer in BT as the optimization target — we believe this last layer choice acts as a regularization to avoid over optimization. We focused on reward modeling instead of an entire DPO workflow to better separate source of gain and for flexibility of downstream RL workflow. Importantly Feng et al. (2025) only choose response conditioning on prompts while we also target on choosing best prompts via joint prompt-response selection.

## 3. Designing of Comparisons

### 3.1. Linear BT Regression.

Consider a simplified case where the true reward function is linear with respect to some intermediate embedding, $r(\mathbf{\Phi}_{i,1}) = \mathbf{\Phi}_{i,1}^{\top}\boldsymbol{\beta}_{-1}$, for weight vector $\boldsymbol{\beta}_{-1}$. We use $\mathbf{\Phi}$ instead of $\mathbf{\Psi}$ because the reward may not be linear with respect to the original embedding $\mathbf{\Psi}$ used in reward modeling, and we wish to avoid confusion. The subscript $-1$ in $\boldsymbol{\beta}_{-1}$ reflects how we will apply these results in practice: $\mathbf{\Phi}$ represents the output before the final linear layer, and $\boldsymbol{\beta}_{-1}$ corresponds to the weight of this last layer. For now, we assume that this linear feature $\mathbf{\Phi}$ is known to us. Note that there is no bias term because linear BT is identified only up to translation.

Under this simplified setting the preference generating process of $i$th pair $h_i$ can be simplified to

$$h_i \sim \text{Bernoulli}[\sigma[(\mathbf{\Phi}_{i,1} - \mathbf{\Phi}_{i,2})^{\top}\boldsymbol{\beta}_{-1}]] \quad (2)$$

It can be observed that this corresponds to a logistic regression, where the covariates are the difference $\mathbf{\Phi}_{i,1} - \mathbf{\Phi}_{i,2}$.

By applying the theory from generalized linear models, we know that the maximum likelihood estimate $\hat{\boldsymbol{\beta}}_{-1}$ is asymptotically Gaussian distributed, with mean $\boldsymbol{\beta}_{-1}$ and covariance matrix $\mathcal{I}^{-1}$, where $\mathcal{I}$ denotes the Fisher information (FI) matrix (see e.g., Shao, 2008, Ch. 4.5.2). For the linear Bradley-Terry model, the FI is

$$\mathcal{I} = \sum_{i=1}^{I} (\mathbf{\Phi}_{i,1} - \mathbf{\Phi}_{i,2})(\mathbf{\Phi}_{i,1} - \mathbf{\Phi}_{i,2})^{\top} p_i(1 - p_i) \quad (3)$$

Where $p_i = \sigma[(\mathbf{\Phi}_{i,1} - \mathbf{\Phi}_{i,2})^{\top}\boldsymbol{\beta}_{-1}]$, it can be observed that $p_i(1 - p_i)$ represents the variance of a Bernoulli random variable.

The Fisher information matrix can be interpreted as the metric tensor in a Riemannian manifold of distributions, where the distance between them is given by the symmetrized KL divergence (Costa et al., 2015). FI quantifies the amount of information in the dataset for estimating the parameters $\boldsymbol{\beta}_{-1}$. From a Bayesian perspective, the Bernstein-von Mises theorem (Van der Vaart, 2000, Ch. 10.2, Thm 10.1) states that $\mathcal{I}^{-1}$ is also the asymptotic covariance matrix of the posterior distribution of $\boldsymbol{\beta}_{-1}$, assuming mild regularity conditions on the prior.

The FI can be viewed as a sum over all independent data points' contribution. For each data point, there are two terms multiplied together: the empirical covariance of embedding differences $(\mathbf{\Phi}_{i,1} - \mathbf{\Phi}_{i,2})(\mathbf{\Phi}_{i,1} - \mathbf{\Phi}_{i,2})^{\top}$, and $p_i(1 - p_i)$, the variance of the comparison results. Sun et al. (2025a) suggested that improving the variance of comparisons can be interpreted as improving annotation quality which can also be seen from FI.

To make the FI large Equation (3) an ideal comparison should exhibit both a large variance in the embedding difference (thus $(\mathbf{\Phi}_{i,1} - \mathbf{\Phi}_{i,2})(\mathbf{\Phi}_{i,1} - \mathbf{\Phi}_{i,2})^{\top}$ having large eigenvalues) and a high variance in the comparison outcomes (thus $p_i(1 - p_i)$ large). This implies that the embedding space should be diverse, such that $\mathbf{\Phi}_{i,1} - \mathbf{\Phi}_{i,2}$ captures a wide range of differences, and each comparison should be informative—not too close to 0 or 1. The former encourages exploration within the embedding space, leading to a better regression model, while the latter ensures that comparisons are not trivial, improving sample efficiency. An everyday analogy for comparing non-obvious pairs would be that comparing a world champion to a newbie in chess offers little insight into the abilities of either player.

The FI plays a crucial role in the classical theory of experimental design, both in frequentist and Bayesian frameworks, as highlighted by the Bernstein-von Mises theorem. This leads to a family of design strategies known as alphabetical designs (Chaloner & Verdinelli, 1995; Pukelsheim, 2006).

**(Bayesian) D-optimality (Chaloner & Verdinelli, 1995).** The alphabetical designs focus on the (co)variance of either estimating weights $\boldsymbol{\beta}_{-1}$ or making predictions under new embeddings, typically summarized through the covariance matrix. For example, the D-optimal design minimizes the determinant of the (asymptotic) covariance matrix of the last layer weights, $\boldsymbol{\beta}_{-1}$. Since $|\mathcal{I}^{-1}| = 1/|\mathcal{I}|$, this is equivalent to maximizing the determinant of the FI.

The Bayesian variant of D-optimal involves having prior contribution, such as maximizing $|\mathcal{I} + I/\sigma^2|$, where $I$ is the identity matrix, to avoid a determinant of zero. This corresponds to the inverse covariance matrix of the Laplace approximation of the posterior of $\boldsymbol{\beta}_{-1}$, assuming a normal prior with variance $\sigma^2$.

A plug-in estimator of $p_i$, $\hat{p}_i$, using the current best model, can be used to estimate the FI (Chaloner & Verdinelli, 1995; Pukelsheim, 2006). In this approach, the scoring rule is the determinant of the Fisher Information matrix.

$$\mathcal{S}_{\text{dopt}}(\mathcal{C}) = |\sum_{i \in \mathcal{C}} (\boldsymbol{\Phi}_{i,1} - \boldsymbol{\Phi}_{i,2})(\boldsymbol{\Phi}_{i,1} - \boldsymbol{\Phi}_{i,2})^\top \hat{p}_i(1 - \hat{p}_i)| \quad (4)$$

In experiments, we refer to this strategy as `D-opt`. Other forms of optimality also exist, each targeting different summaries of the Fisher Information (FI), such as A-optimality, which focuses on minimizing the trace of $\mathcal{I}^{-1}$. When the prediction of a new, known embedding is the primary concern, G-optimality aims to minimize the variance of predictions on new embeddings.

In this work, we suggest using D-optimality because it avoids the need to invert the FI, as required in A-optimality, and doesn't require specifying which samples to predict, as in G-optimality. For readers interested in further details, we refer to Pukelsheim (2006) (Ch. 9).

The D-optimality strategy can be made a past-aware version by incorporating previously collected data. The asymptotic covariance of the full data-conditioned posterior is then $(\mathcal{I}_{\text{past}} + \mathcal{I})^{-1}$, where $\mathcal{I}_{\text{past}}$ is computed using prior data and Equation (3). This approach relates to Bayesian methods like Bayesian active learning by disagreement (BALD) (Houlsby et al., 2011), which minimizes posterior entropy. Since Gaussian entropy is proportional to the log-determinant of its covariance. In our experiments, we refer to this variant as `PA D-opt`.

Next, we review some other strategies that can be applied to BT models.

**Entropy Sampling (Settles, 2009; Muldrew et al., 2024).** This strategy aims to select samples about which the current model is most uncertain (Settles, 2009). In the context of binary preference modeling, this corresponds to choosing data whose predictions $\hat{p}_i$ are closest to 0.5, effectively exploring the level set of the reward. This is similar to a binary classification problem where the goal is to explore the decision boundary. This approach was also proposed by Muldrew et al. (2024) as maximizing predictive entropy. The scoring rule is then,

$$\mathcal{S}_{\text{entropy}}(\mathcal{C}) = \sum_{i \in \mathcal{C}} [-\hat{p}_i \log \hat{p}_i - (1 - \hat{p}_i) \log(1 - \hat{p}_i)] \quad (5)$$

Since the entropy of a Bernoulli distribution reaches its maximum when $p = 0.5$, this approach is equivalent to selecting the top $c$ pairs where the predicted probability is closest to 0.5. In our experiments, we refer to this method as `Entropy`.

**Maximum Difference (Muldrew et al., 2024).** Contrasting with entropy sampling, this strategy focuses on comparing

samples that the current reward model predicts to be the best and the worst, corresponding to probabilities close to 0 or 1. This approach was used by Muldrew et al. (2024) to measure model certainties. The scoring rule to be maximized can thus be interpreted as difference in estimated reward $|\hat{r}_{i,1} - \hat{r}_{i,2}|$.

$$\mathcal{S}_{\text{maxdiff}}(\mathcal{C}) = \sum_{i \in \mathcal{C}} |\hat{r}_{i,1} - \hat{r}_{i,2}| \quad (6)$$

This strategy encourages exploration in the *reward* space rather than the embedding space. It is sometimes used in active learning when the goal is to identify positive examples rather than the best classification (Settles, 2009). This justifies its use in reward modeling, where the goal is to obtain responses that yield better rewards in downstream tasks. In our experiments, we refer to this method as `Maxdiff`.

**Optimizing Design Matrix (Mukherjee et al., 2024).** This strategy focuses on finding the best collection of embeddings, or the design matrix in statistics terms $\boldsymbol{\Phi}_{i,1} - \boldsymbol{\Phi}_{i,2}$, without looking at model predictions. A common objective is to optimize the covariance matrix of the designs, $\Sigma = \sum_{i=1}^I (\boldsymbol{\Phi}_{i,1} - \boldsymbol{\Phi}_{i,2})(\boldsymbol{\Phi}_{i,1} - \boldsymbol{\Phi}_{i,2})^\top$. One approach is to maximize the determinant of $\Sigma$, $|\Sigma|$, which encourages exploration over a large space of embedding differences. In fact, if we assume a linear regression model with additive Gaussian noise instead of logistic regression, this covariance matrix corresponds to the Fisher Information matrix of the regression coefficients, and this strategy aligns with the D-optimal design. The scoring rule is

$$\mathcal{S}_{\text{XtX}}(\mathcal{C}) = |\sum_{i \in \mathcal{C}} (\boldsymbol{\Phi}_{i,1} - \boldsymbol{\Phi}_{i,2})(\boldsymbol{\Phi}_{i,1} - \boldsymbol{\Phi}_{i,2})^\top| \quad (7)$$

Mukherjee et al. (2024) used a similar strategy for a different type of preference data that is not purely binary. In our experiments, we refer to this method as `det(XtX)`, for the determinant of $X^\top X$.

**Coreset (Huggins et al., 2016; Munteanu et al., 2018).** Instead of minimizing uncertainty in parameter estimations, the Coreset strategy aims to find a small subset of samples such that the trained model closely approximates the one trained on the full dataset, effectively transforming the problem into a sparse approximation task on weighting data points. The Coreset method for logistic regression has been studied recently by Munteanu et al. (2018) and Huggins et al. (2016) in both frequentist and Bayesian settings. In our experiment, we adopted the method of Huggins et al. (2016). The scoring rule does not have a simple closed-form solution, so we refer interested readers to Huggins et al. (2016) and denote it as $\mathcal{S}_{\text{coreset}}$. In our experiments, we refer to this method as `Coreset`.

**BALD and batchBALD (Houlsby et al., 2011; Kirsch et al., 2019).** When transitioning from frequentist to Bayesian framework, BALD (Houlsby et al., 2011) and

BatchBALD (Kirsch et al., 2019) select data with high mutual information between the candidate batch's prediction and model parameters, making the data more informative. Houlsby et al. (2011) showed that this approach maximizes expected posterior entropy reduction. This strategy applies to preference learning (Houlsby et al., 2011) but requires a Bayesian model. We denote the corresponding scoring rule as $\mathcal{S}_{\text{bBALD}}$. In our experiments, we refer to this method as `BatchBald`. We used implementation in `batchbald_redux` (Kirsch et al., 2019).

This strategy relates to Bayesian D-optimality; when posterior entropy is tractable, it can be minimized directly instead of relying on approximations from Houlsby et al. (2011). If the posterior is Gaussian, entropy is proportional to the log-determinant of its covariance, leading to D-optimality.

### 3.2. Gradient Approximation for Combinatorial Optimization.

In some strategies, we select a data subset to maximize information criteria like the determinant of FI or the design matrix. These often lead to intractable combinatorial optimization problems. To address this, we use the sensitivity approach from the coreset and robustness literature (Huggins et al., 2016; Campbell & Broderick, 2018; 2019). When the information criteria are expressed as a nonlinear function over sum of data point contributions, i.e., $\mathcal{S} = f(\sum_i c_i)$, where each data point contributes $c_i$, we introduce weights $w_i$, allowing the score to be rewritten as $\mathcal{S}(\boldsymbol{w}) = f(\sum_i w_i c_i)$. For instance, the D-optimal score expresses the determinant of FI of a subset $\mathcal{C}$ as a weighted sum.

$$\mathcal{S}_{\text{dopt}}(\boldsymbol{w}) = |\sum_i w_i (\boldsymbol{\Phi}_{i,1} - \boldsymbol{\Phi}_{i,2})(\boldsymbol{\Phi}_{i,1} - \boldsymbol{\Phi}_{i,2})^\top \hat{p}_i (1 - \hat{p}_i)| \quad (8)$$

Each candidate pair is assigned a weight $w_i = 1_{i \in \mathcal{C}}$. Selecting a subset $\mathcal{C}$ to maximize $\mathcal{S}_{\text{dopt}}$ is equivalent to finding a sparse 0-1 weight vector $\boldsymbol{w}$ that maximizes $\mathcal{S}_{\text{dopt}}(\boldsymbol{w})$.

To approximate the optimization, we treat $\boldsymbol{w}$ as continuous and perform a Taylor expansion around $\boldsymbol{w} = \mathbf{1}$, the all 1 vector, i.e., all data points are included.

$$\mathcal{S}(\boldsymbol{w}) \approx \mathcal{S}(\mathbf{1}) - (\mathbf{1} - \boldsymbol{w})^\top \nabla_{\boldsymbol{w}} \mathcal{S}(\boldsymbol{w})|_{\boldsymbol{w}=\mathbf{1}} \quad (9)$$

The approximated optimization problem becomes

$$\operatorname{argmax}_{\boldsymbol{w}} \mathcal{S}(\boldsymbol{w}) \approx \operatorname{argmax}_{\boldsymbol{w}} \boldsymbol{w}^\top \nabla_{\boldsymbol{w}} \mathcal{S}(\boldsymbol{w})|_{\boldsymbol{w}=\mathbf{1}} \quad (10)$$

A sparse 0-1 valued vector $\boldsymbol{w}$ that optimizes the right-hand side of Equation (9) can be obtained by selecting the data points with the largest gradient, $\nabla_{\boldsymbol{w}} \mathcal{S}(\boldsymbol{w})|_{\boldsymbol{w}=\mathbf{1}}$. A probabilistic approach, when all gradients are positive, involves sampling according to the weights given by $\nabla_{\boldsymbol{w}} \mathcal{S}(\boldsymbol{w})|_{\boldsymbol{w}=\mathbf{1}}$.

### 3.3. Handling nonlinear model using last layer features.

For nonlinear reward models in Equation (2), the dependencies on embeddings become more complex. Strategies like maximum difference and entropy sampling, which depend only on model predictions, remain unaffected by the architecture, while batchBALD is designed for (Bayesian) deep models. Feature-based methods like coreset or D-optimal need adaptation. A heuristic from the Bayesian last layer (Tran et al., 2019) and computer vision literature Sener & Savarese (2017) suggests using the last layer before the linear output as a feature, applying linear strategies to it.

$$r(\boldsymbol{\Psi}) = F_{\boldsymbol{\theta}}(\boldsymbol{\Psi})^\top \boldsymbol{\beta}_{\text{-1}} \quad (11)$$

For some nonlinear function $F_{\boldsymbol{\theta}}$ parameterized by $\boldsymbol{\theta}$, e.g., an MLP and $\boldsymbol{\Phi} := F_{\boldsymbol{\theta}}(\boldsymbol{\Psi})$. We apply methods in linear settings with features $F_{\boldsymbol{\theta}}(\boldsymbol{\Psi})$. We then train $\boldsymbol{\theta}$ and $\boldsymbol{\beta}_{\text{-1}}$ together once data are labeled. In particular, in Sener & Savarese (2017) the nonlinear function $F_{\boldsymbol{\theta}}$ is a CNN and they took a coreset approach. Here we apply this strategy to the coreset, optimal design matrix and D-optimal setting.

## 4. Illustrative Examples in Dimension Two

**Experiment Setups** In this experiment, we provide a two-dimensional example of the comparisons made by each strategy. The ground truth reward was defined as the log probability of a mixture of two Gaussians, centered at $(-2.5, -2.5)$ and $(2.5, 2.5)$ with a variance of 0.25. Preference data was simulated using the BT model, and we attempted to learn the reward function with a 3-layer MLP with 16 hidden units. For each round, 1000 points were sampled from a standard normal distribution, and 200 comparisons were selected using different strategies. 4 rounds are shown in Figure 1, with a zoomed-in version in Figure 6 and performance measured by 1-Spearman correlation in Figure 7. D-optimal method outperform alternatives including full Bayesian method like BatchBALD.

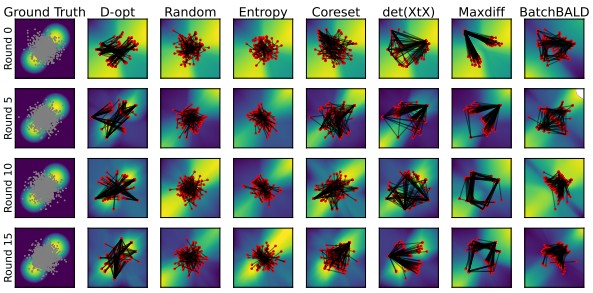

Figure 1: Comparisons drawn by different strategies to learn a 2D bimodal reward function. The heat map showed the estimated functions. Red dots connected by lines are **selected pairs** and gray dots on the first column are candidate points to choose from.

**What were compared in dimension two?** We observed that D-optimal selects diverse samples with many anchoring points, often comparing multiple points to a single one,

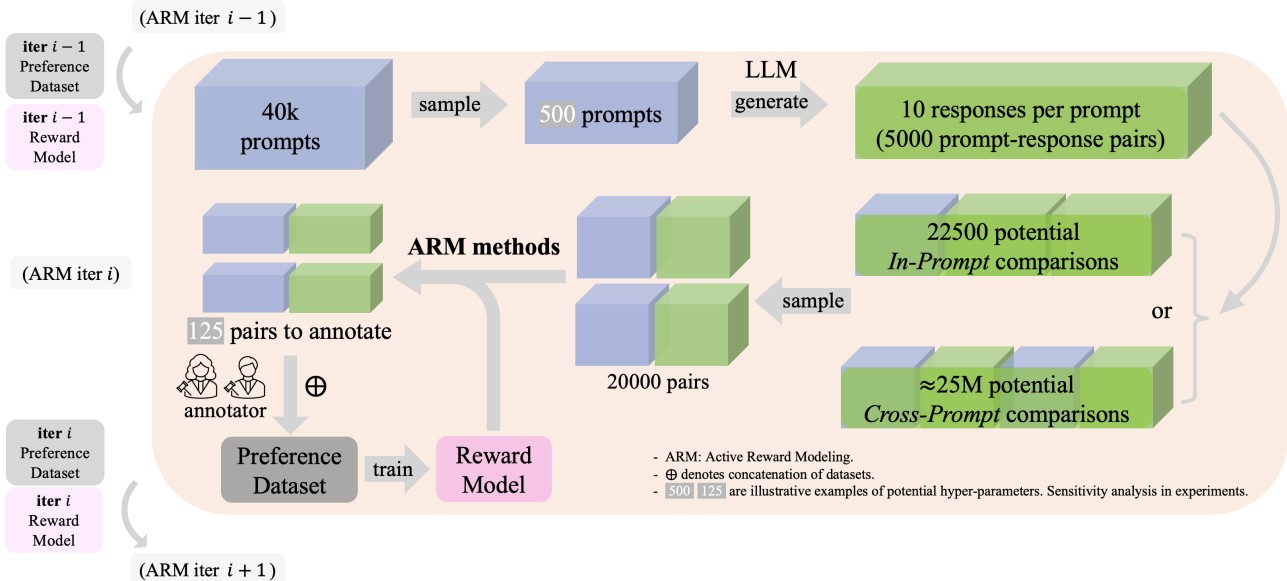

Figure 2: Workflow of active reward modeling and experimental setups. At each round, we start with randomly sampling prompts, generating responses and candidate comparisons, active labeling, and model retraining.

spreading out the level set in the original space. Entropy sampling, similar to random sampling, focuses on points near reward values, effectively traversing the reward function's level set. Coreset also selects diverse comparisons, though not always among points with similar reward values. The best design matrix method behaves similarly to coreset, emphasizing diversity in comparisons. In contrast, the max difference method tends to compare extreme values with many others, promoting exploration but potentially yielding less informative comparisons. BatchBALD also selects diverse comparisons, though without a clear pattern. These observations suggest that most methods encourage exploration, entropy sampling prioritizes informative comparisons, and D-optimal seeks a balance between the two.

## 5. Experiments with LLMs

### 5.1. Overall Setup

In this section, we test different design strategies in LLM applications. We start with discussions of general experiment setups and the evaluation metrics we used in experiments.

**Objective and Evaluation Metrics.** We assess the data efficiency of various comparison selection methods. The main metrics are $1-$ Spearman's rank correlation and best-of-N test-time reward, as reward modeling aims to order responses correctly and select the best one during test time. The golden reward models from Dong et al. (2024) serve as the surrogate for ground truth. Specifically, we consider

- **Batched Spearman's Correlations**: we measure the ranking correlation within each test prompt across 500 generations (Sun et al., 2025a). We took $1-$ Spearman's

correlations as a test set metric.
- **Best-of-N Reward**: we evaluate the best-of-N (N=500) reward on test prompts (Gao et al., 2023; Gui et al., 2024).

A method is considered superior if it achieves a smaller $1-$ Spearman's correlation, a larger Best-of-N reward, or the same performance with fewer annotations.

**Base Models, Annotations, and Golden Reward Models.** We conducted experiments using three open-source LLMs: Gemma2b, Gemma7b, and LLaMA3-8b (Team et al., 2024; Meta, 2024). To ensure affordability, we followed methods from Gao et al. (2023); Liu et al. (2023); Tran & Chris Glaze (2024); Dong et al. (2024); Sun et al. (2025a) to use open-source golden reward models as annotators. We used the Anthropic `Harmless` and `Helpful` datasets (Bai et al., 2022a) that has been widely studied in reward modeling, and golden reward models are available (Yang et al., 2024; Dong et al., 2023; 2024). The dataset includes 40k prompts with 10 responses each for training, and 2k prompts with 500 generations each for testing.

**Reward Modeling.** To separate representation learning from reward modeling, we train our reward model using joint embeddings of prompts and responses. An MLP with three hidden layers and BT loss was used. Since the BT model is not identified up to a translation, we exclude bias in the final linear layer. Our ablation studies show that the size of hidden units does not significantly affect the results. For more details, see Appendix A.5.

**Online Annotation Pipeline.** We train our model sequentially, increasing the sample size at each step. At the beginning of each step, we randomly draw 500 prompts. For

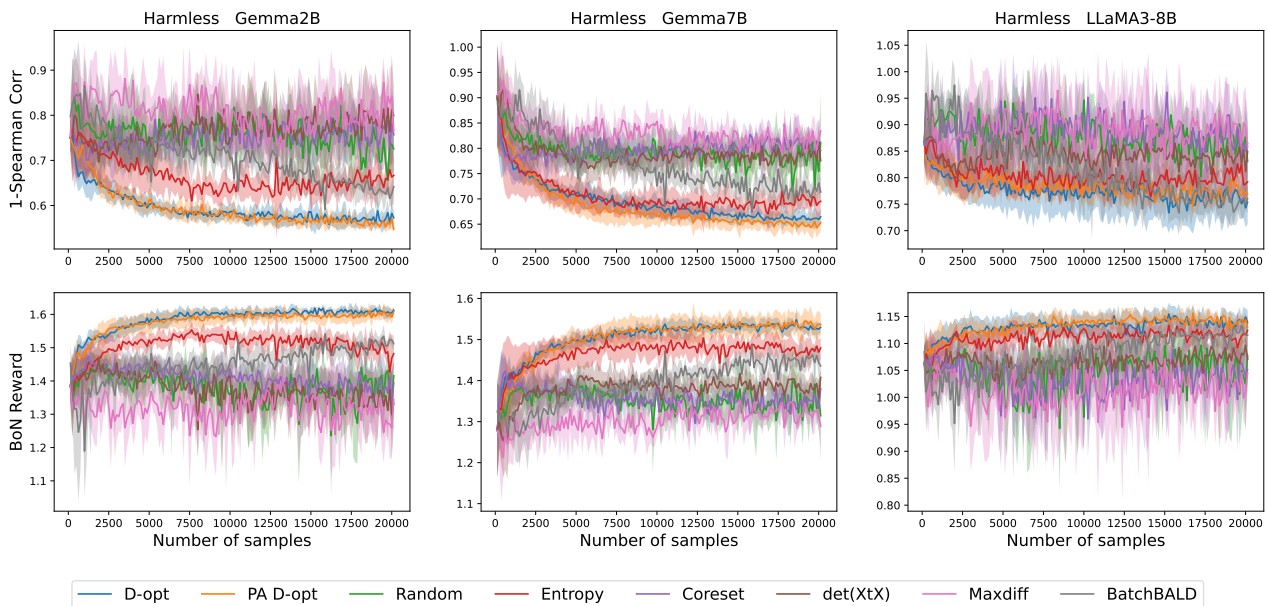

Figure 3: Comparing annotation efficiency of different methods. (`Harmless` Dataset, 3 Models, 8 Methods). First row: 1− Spearman's Correlation (**lower is better**); second row: Best-of-N reward (**higher is better**). Experiments are repeated with 5 seeds.

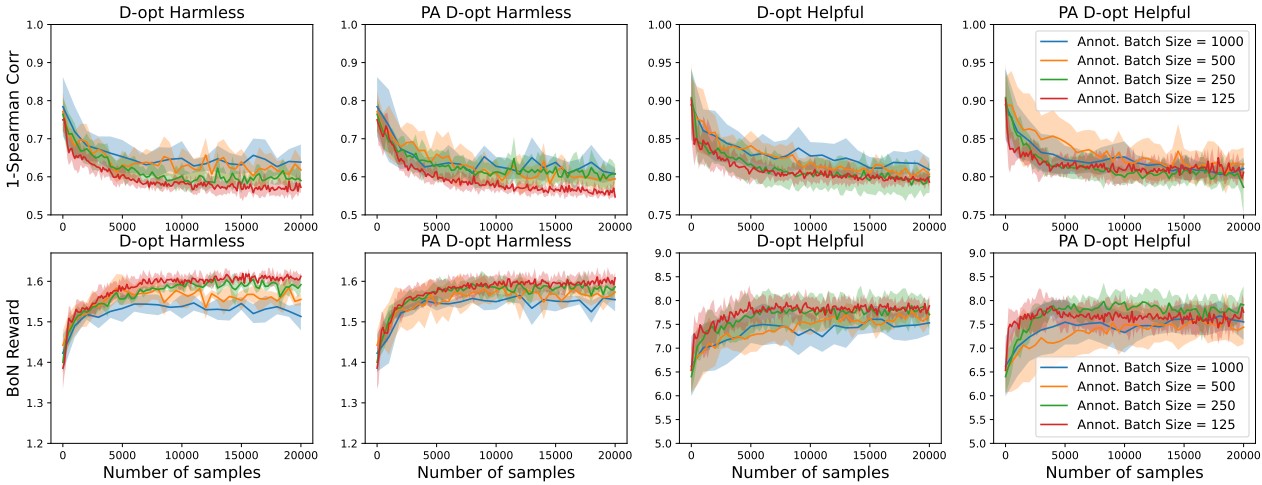

Figure 4: Investigating how annotation batch size choices affect learning performance of our methods. Model: Gemma 2B. The first two columns present results on the `Harmless` dataset, and the second two columns present results on the `Helpful` dataset. First row: 1− Spearman's Correlation (**lower is better**); second row: Best-of-N reward (**higher is better**). The results presented are from 5 runs with different seeds.

each of the 500 prompts, we randomly select 2 out of 10 responses for in-prompt comparisons, yielding $500 \times \binom{10}{2} = 22,500$ potential comparisons. Note that since these $22,500$ contains repeated use of prompt-response pairs, therefore needs only $500 \times 10 = 5,000$ generations rather than $22,500$ generations. For cross-prompt annotations, there are approximately 25 million potential comparisons. We randomly sample a fix-sized subset 20000 out of those potential comparisons for different algorithms to choose from, see Figure 2.

At each online iteration, strategies that require model pre-

dictions use the reward model from the previous iteration. We test different `annotation batch sizes`, an important hyperparameter to tune, ranging from $125, 250, 500, 1000$ to understand performance across various settings. After annotation, we retrain the entire model and evaluate it after each re-training.

### 5.2. Comparing Annotation Efficiency

Figure 3 presents results on the `Harmless` dataset (see Appendix A.2, Figure 8 for `Helpful` results). **D-opt** and **Past-Aware D-opt** outperform other methods, demonstrat-

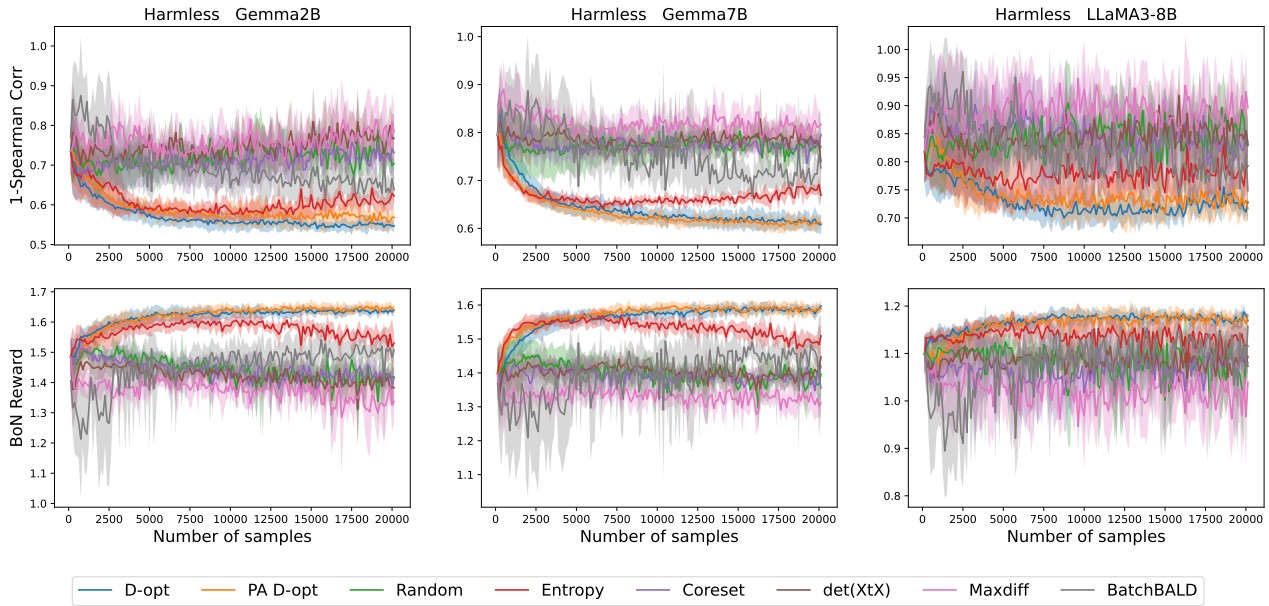

Figure 5: Comparing annotation efficiency of different methods under the **Cross-Prompt** annotation setups. (`Harmless` Dataset, 3 Models, 8 Methods). First row: 1− Spearman's Correlation (**lower is better**); second row: Best-of-N reward (**higher is better**). Experiments are repeated with 5 seeds.

ing both superior performance and greater stability. In contrast, alternative approaches exhibit training instability and significantly higher variance during online learning.

### 5.3. Comparing Annotation Batch Sizes

In this section, we evaluate different methods under varying `annotation batch size` setups, ranging from 125 to 1000. Notably, our proposed methods are computationally efficient: since the reward model operates on embeddings, re-training a 3-layer MLP with 10k annotations takes only a few minutes on a GPU server—while human annotation is significantly more time-consuming.

Figure 4 presents results for the Gemma2B model (results for the other two base models are in Appendix A.3 due to space constraints). Overall, **D-opt** and **Past-Aware D-opt** consistently outperform other methods across different annotation batch sizes. Additionally, we observe performance improvements when using smaller batch sizes, corresponding to a more online setup. Given the low computational cost, this suggests a practical strategy: using small annotation batches with frequent model retraining to enhance annotation efficiency in reward model development.

### 5.4. Results with Cross-Prompt Annotations

Cross-prompt annotation has been proposed as an alternative to in-prompt comparison, demonstrating superior performance in LLM alignment tasks (Yin et al., 2024; Sun et al., 2025a, see also Appendix B). To assess the generality of our methods, we extend our experiments to cross-prompt

setups and compare different approaches.

Figure 5 shows the results under cross-prompt annotation. **D-opt** and **Past-Aware D-opt** achieve significantly better performance in both annotation efficiency and alignment across tasks and base models.

Comparing Figure 5 with Figure 3, we observe efficiency gains across all methods, with the entropy-based approach exhibiting the most substantial improvement. Appendix A.4 provides a direct comparison between in-prompt and cross-prompt annotations for interested readers.

### 5.5. Hyper-Parameter Sensitivity Analysis

To examine the sensitivity of different methods to hyper-parameter choices and provide insights for real-world applications, we varied two key factors: **Candidate Number** and **Hidden Dimension of Reward Model MLPs** across different active reward modeling designs.

Our sensitivity analysis reveals that all algorithms remain robust and are largely insensitive to specific hyper-parameter choices in our embeddings-as-inputs setup. Detailed results are provided in Appendix A.5.

## 6. Discussion

**Designing Comparisons.** Our experiments show that applying the classic method to the last-layer features yields strong performance and stability. The D-opt method is also highly efficient, as its information criteria and optimization procedure are largely analytical, enabling real-time pair se-

lection. This is valuable when collecting user preferences in a user interface without introducing significant latency. Additionally, this approach might be adapted to other model architectures, including e.g., vision-language models.

**An Empirical Bayes View and Stability of Last-Layer Design.** The connection between the last-layer D-optimal method and BALD can be seen by considering previous layers as a transformation of the Gaussian prior for the last layer's weights. These previous layer weights act as hyper-parameters of the prior, which are fitted using maximum likelihood, akin to an empirical Bayes procedure (Deely & Lindley, 1981). By minimizing posterior entropy, we perform D-optimal design followed by Gaussian approximation after the transformation. Empirical Bayes helps reduce the subjectivity and uncertainty in selecting priors, potentially explaining the robustness of our method compared to full Bayesian approaches like batchBALD, which involve hyper-priors on these hyperparameters.

**Classic Experimental Design Methods in the Foundation Model Era.** We conjecture that the success of using the last layer in classical experimental design stems from the fact that the embeddings are already close to linear features. Given the extensive study of experimental design in generalized linear models (see e.g., Atkinson et al., 2007; Pukelsheim, 2006), we believe it is a general strategy to apply these methods to the last layer of deep learning models, particularly when leveraging learned representations from foundation models.

**Comparison of Reward Model Training Cost, Annotation Cost, and Response Generation Cost** In this section, we highlight the relationship among the cost of the reward model training, annotation, and LLM (multiple) response generation. In most cases

$$\text{cost}_{annotation} \gg \text{cost}_{generation}$$

Specifically, while our method requires generating more responses per prompt, the cost is modest. For 500 prompts, generating 10 responses per prompt instead of 2 increases the generation cost by approximately \$2.5 (from \$0.7 to \$3.2 in total, using DeepSeek API at off-peak rates). However, the number of possible pairwise comparisons grows quadratically with the number of responses — for example, from 1 to $\binom{10}{2} = 45$ comparisons per prompt — leading to up to **45× more training signal** per prompt.

This enables more efficient use of the annotation budget. Assuming a fast annotation speed of 125 comparisons per hour, annotating 20,000 randomly selected comparisons would take 160 hours, costing over \$1,000 at the U.S. federal minimum wage (\$7.25/hour). Our method achieves similar reward model performance using only approximately 1,000 informative comparisons (see Fig. 3), costing less than \$50 — a **20× reduction in annotation cost**.

In addition, if we train embedding based reward models (e.g., BT with pretrained embeddings Sun et al., 2025a;b), it is computationally light. For a reward model with $d$-dimensional input, $n$ comparisons, and $E$ training epochs, total computational cost is $\mathcal{O}(nEd)$ FLOPs. This is negligible compared to LLM response generation — e.g., generating two 1000-token responses per prompt using a 7B LLM may require $\sim 10^{14}$ FLOPs — and drastically cheaper than human labeling.

**Limitations and Future Directions.** Our proposed active learning scheme relies on a well-trained embedding model. The effectiveness of selecting comparisons based on last-layer features depends on these features being informative, which might in turn requires signal-rich embeddings with low noise as input of the reward model (which is MLP in our experiment). An interesting question is whether embeddings that better capture human values (and thus improve reward modeling) differ fundamentally from those optimized for generation. A related consideration is whether reward modeling in LLMs should start from embedding or earlier. Our method relies on BT model as the main statistical model to learn human preference. This is not necessarily, e.g., one can consider quantile regression (Dorka, 2024) and other type of models beyond binary preference data. The main idea of using last layer feature, treating the statistical model as a (generalized) linear model and apply classic theory on experimental design literature can be used as a general recipe for active reward modeling.E.g., for quantile regression, one can use influence function to calculate the asymptotic variance (Koenker, 2005) as the metric of informativeness of a pair of comparison. This calculation leads to a method similar to optimizing design matrix (Mukherjee et al., 2024).

## Impact Statement

Our work advances the efficiency of aligning LLMs with human values by optimizing preference annotation strategies. Since human feedback is costly and time-consuming, our approach can potentially reduce wasted effort on uninformative comparisons, maximizing the value of each annotation. By improving the efficiency of learning from human preferences, this research has the potential to accelerate the development of safer and more helpful AI systems.

## Software and Data

Code and embeddings to (efficiently) reproduce all results, including all baselines, can be found through `https://github.com/YunyiShen/ARM-FI`. Embeddings for reward modeling and efficient Best-of-N testing are available at `https://github.com/holarissun/embedding-based-llm-alignment`.

## Acknowledgements

YS acknowledge the MIT SuperCloud and Lincoln Laboratory Supercomputing Center for providing HPC resources that have contributed to the research results reported within this paper. YS and HS acknowledge Yaqi Duan (NYU) for discussions in the early stage of this work.

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

# A. Additional Experiment Results

## A.1. Two dimensional example

In this section we provide an enlarged version of figure Figure 1 and the performance measured by Spearman's rank correlation on an 2D grid.

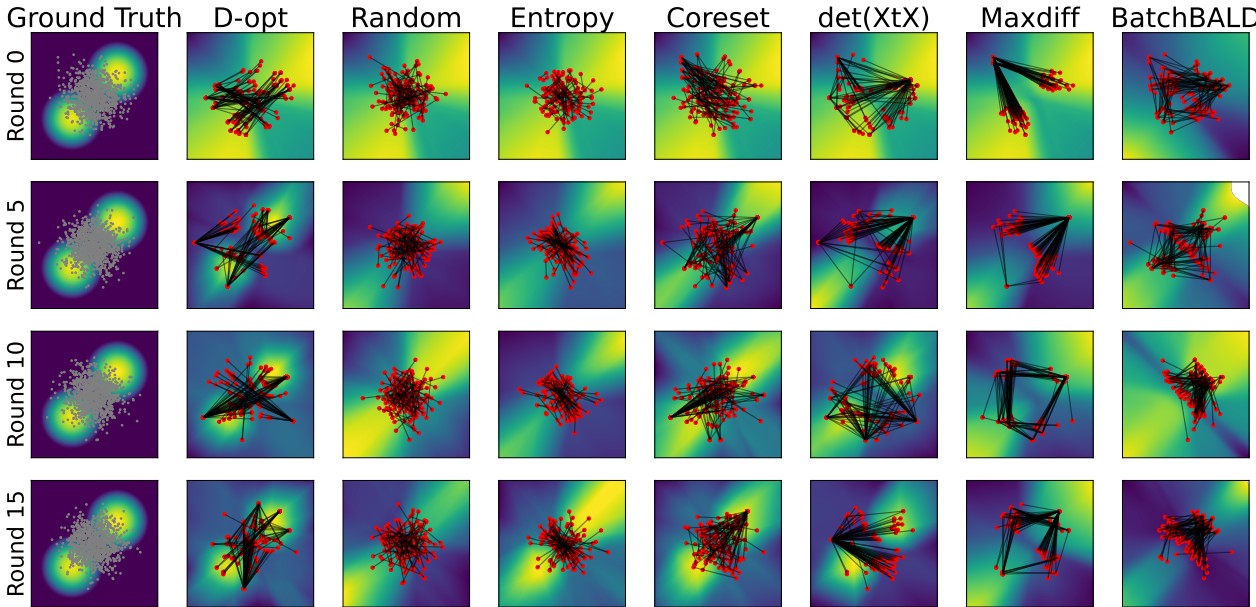

Figure 6: Comparisons drawn by different strategies to better learn a 2D bimodal reward function. The heat map showed the current estimated function. Red crosses connected by black lines are compared pairs and gray dots on the left-most penal are candidate points to be compared.

In Figure 7 we showed the performance measured by 1-Spearman correlation on the entire grid. Last layer D-optimal method outperform other baselines including full Bayesian approach like BatchBALD.

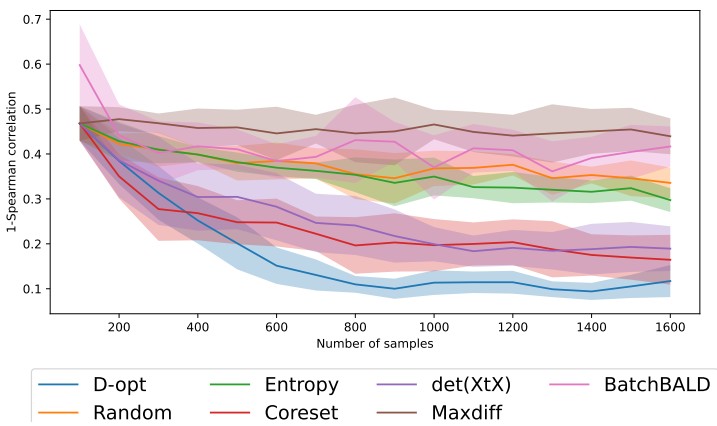

Figure 7: Performance measured by 1-Spearman correlation (smaller the better) in the 2D illustrative example.

## A.2. Comparing Annotation Efficiency on the `Helpful` Dataset

**In-Prompt Annotation** efficiency is provided in Figure 8 (as supplementary of Figure 3 in the main text).

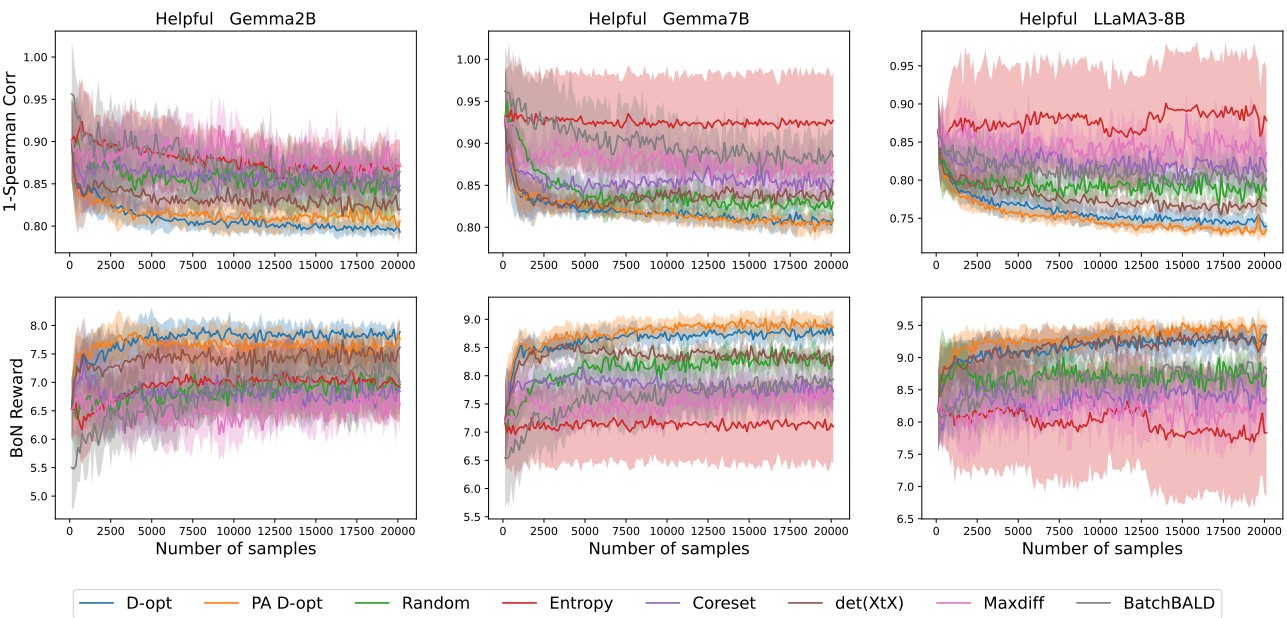

Figure 8: Comparing annotation efficiency of different methods. (`Helpful` Dataset, 3 Models, 8 Methods). First row: 1 - Spearman's Correlation (lower is better); second row: Best-of-N reward. Experiments are repeated with 5 seeds.

**Cross-Prompt Annotation** efficiency is provided in Figure 9 (as supplementary of Figure 5 in the main text).

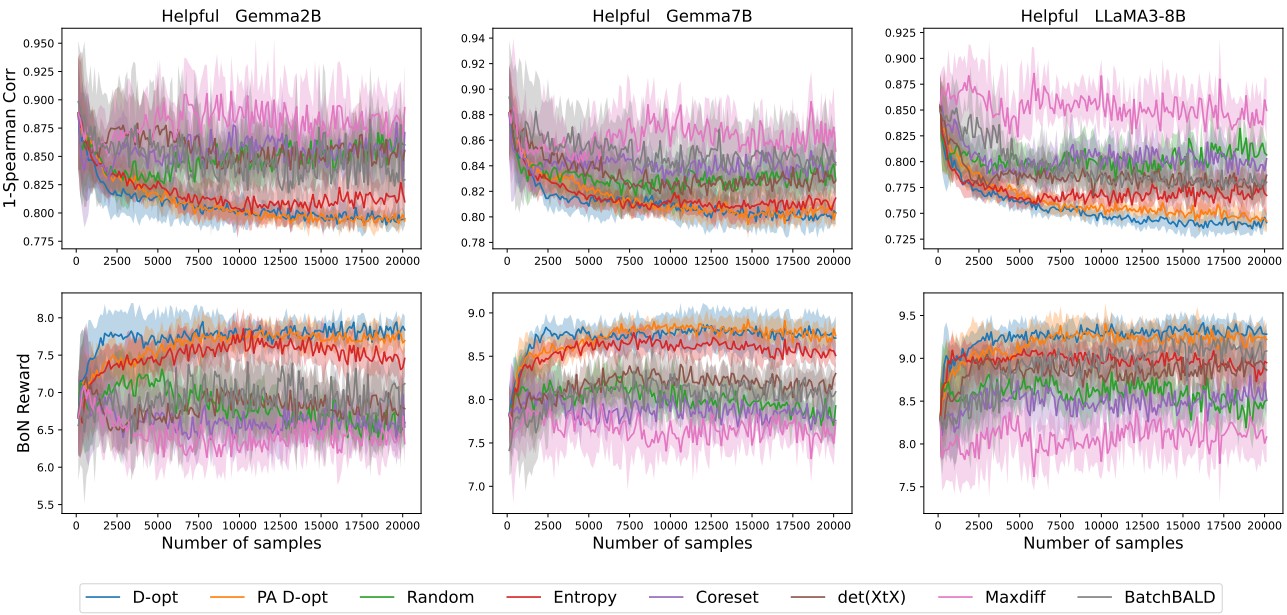

Figure 9: Comparing annotation efficiency of different methods under the **Cross-Prompt** annotation setups. (`Helpful` Dataset, 3 Models, 8 Methods). First row: 1 - Spearman's Correlation (lower is better); second row: Best-of-N reward. Experiments are repeated with 5 seeds.

### A.3. Annotation Batch Size

**Results on All Models**    Due to the space limit of the main text, we deferred the experiment results with Gemma7B and the LLaMA3-8B model when studying the effect of different annotation batch sizes in the following Figures (Figure 10, Figure 11, Figure 12). To summarize the main takeaways — we observe the same trend as we have observed with the Gemma2B model, the proposed methods achieve better performances in the small batch size setups (more online setups). The stability of small batch setups is in general higher than the large batch setups.

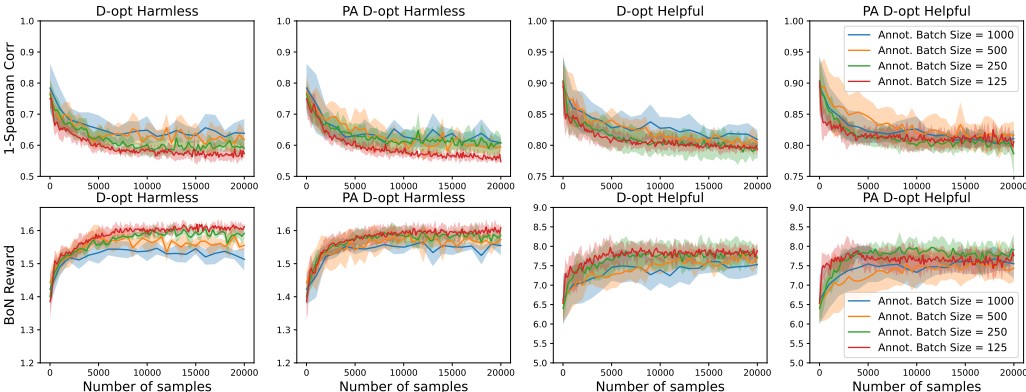

Figure 10: Investigating how annotation batch size choices affect learning performance of different methods. Model: Gemma 2B.

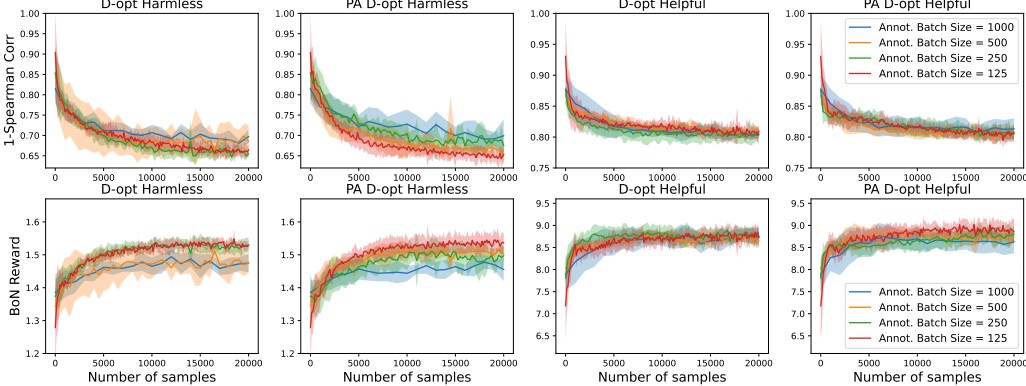

Figure 11: Investigating how annotation batch size choices affect learning performance of different methods. Model: Gemma 7B.

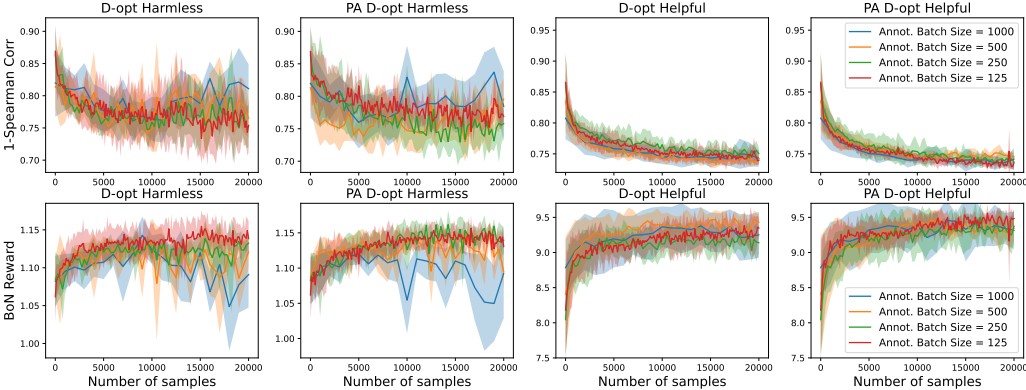

Figure 12: Investigating how annotation batch size choices affect learning performance of different methods. Model: LLaMA3-8B.

**Results with All Methods.** In addition, we use the figures below (Figure 13, Figure 14, Figure 15) for a full analysis on the annotation batch size choices for all methods. For other methods, we do not observe a clear trend on the effect of increasing or decreasing annotation batch sizes.

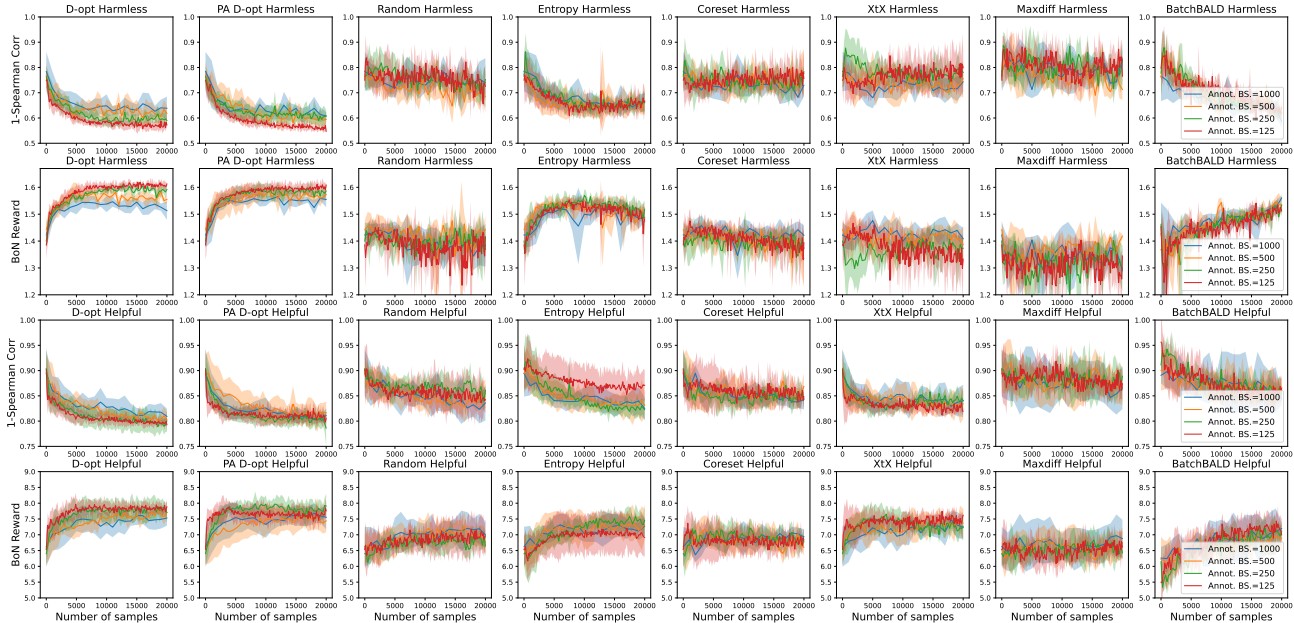

Figure 13: Investigating how annotation batch size choices affect learning performance of different methods. Model: Gemma 2B.

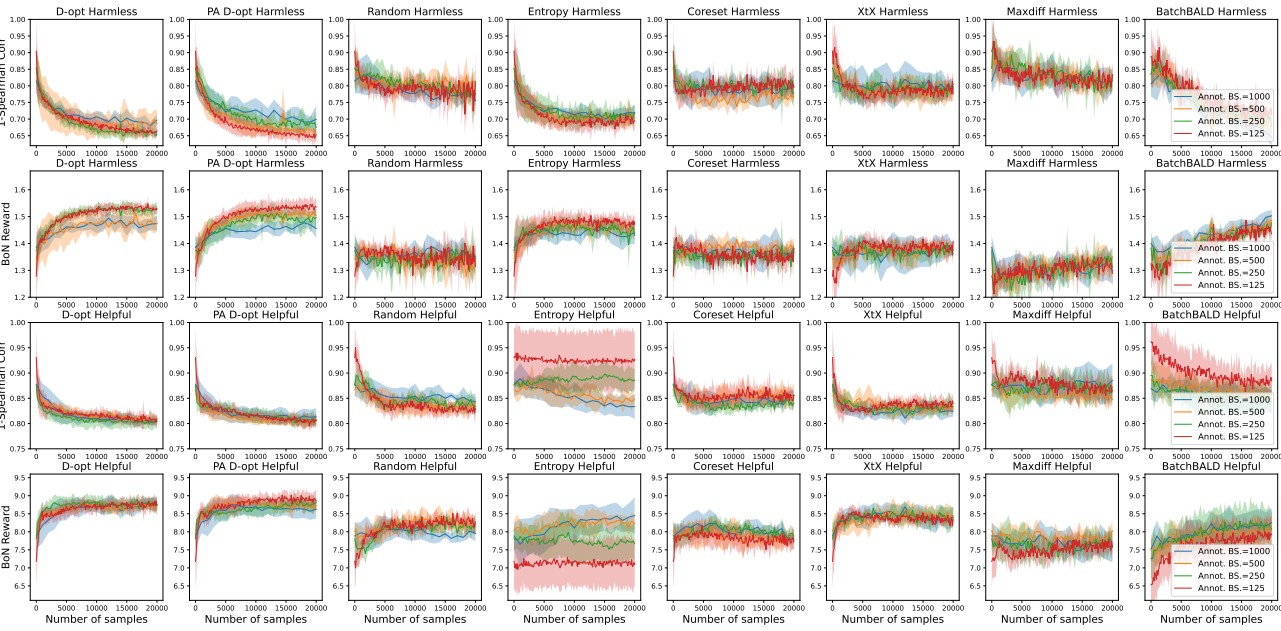

Figure 14: Investigating how annotation batch size choices affect learning performance of different methods. Model: Gemma 7B.

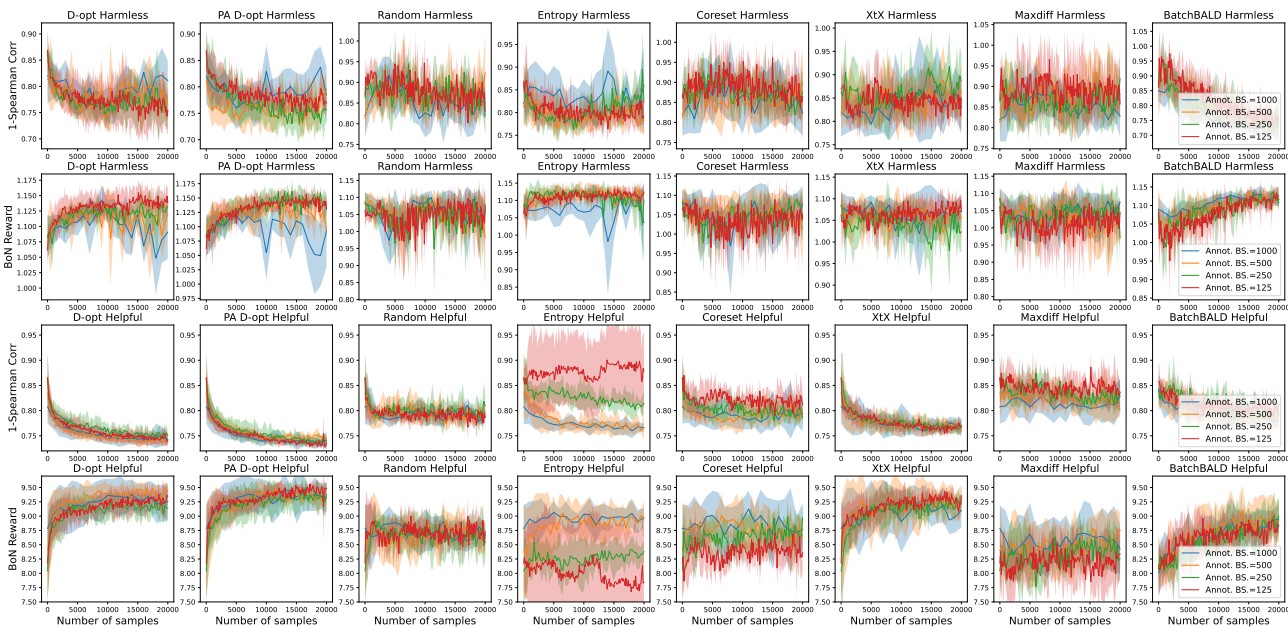

Figure 15: Investigating how annotation batch size choices affect learning performance of different methods. Model: LLaMA3 8B.

## A.4. Compare Cross-Prompt Comparisons and In-Prompt Comparisons

In this section, we provide direct comparisons of learning efficiency when using cross-prompt annotations and in-prompt annotations. In most cases, annotating comparisons using cross-prompt comparison improves learning efficiency, and this can be observed across all methods. Specifically, with the entropy-based method, cross-prompt annotations bring a noticeable boost to learning efficiency and reward model performance.

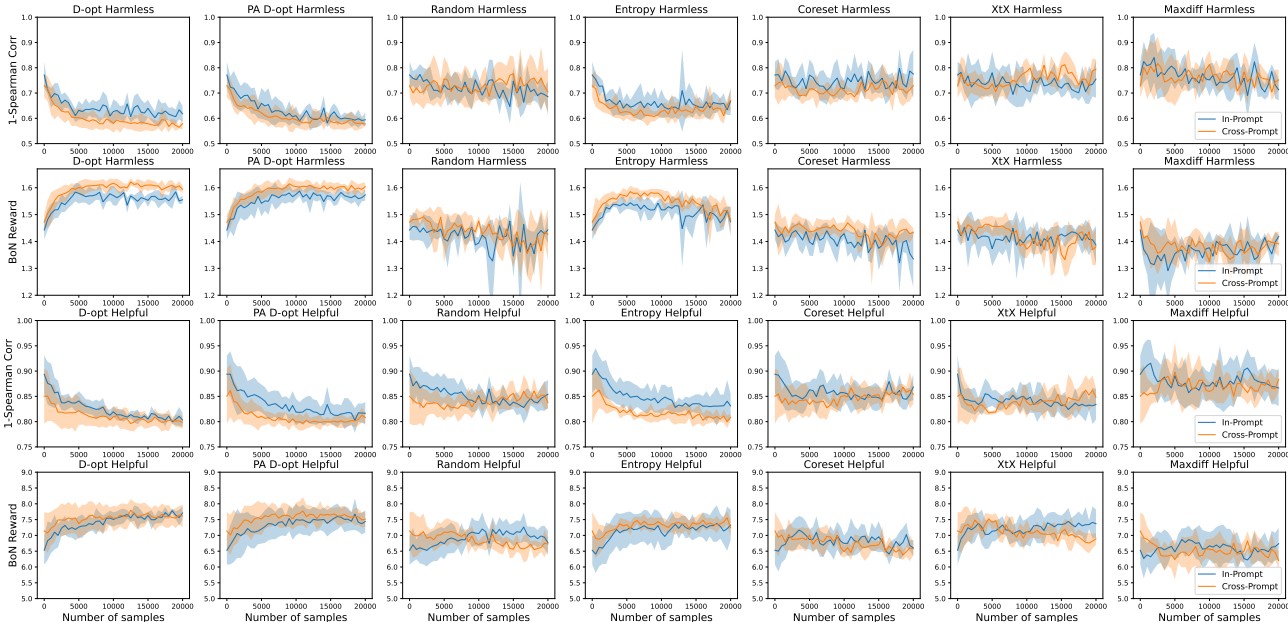

Figure 16: Cross-Prompt preference annotation improves overall annotation efficiency. Annotation batch size 500. Model: Gemma2B.

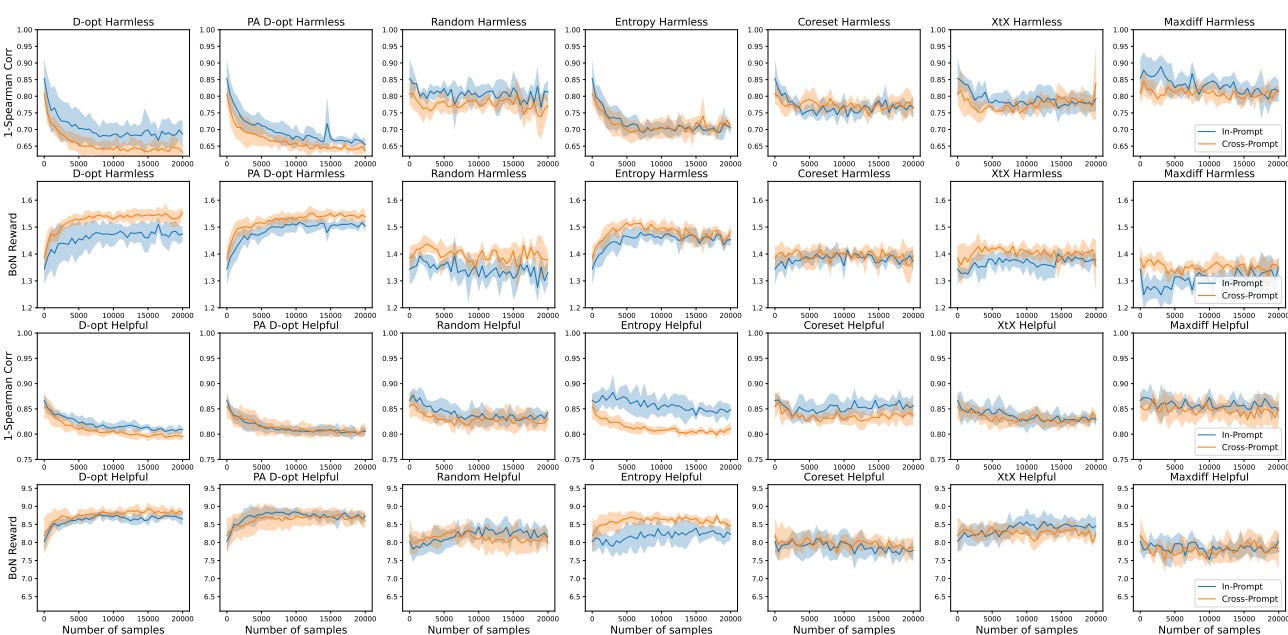

Figure 17: Cross-Prompt preference annotation improves overall annotation efficiency. Annotation batch size 500. Model: Gemma7B.

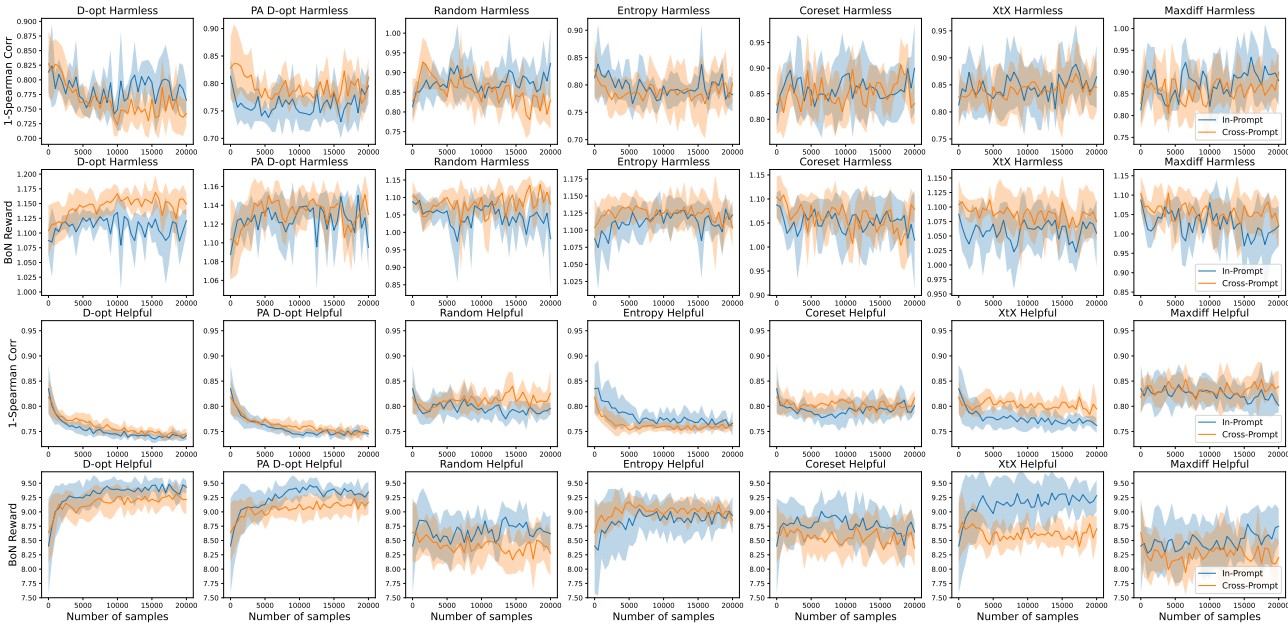

Figure 18: Cross-Prompt preference annotation improves overall annotation efficiency. Annotation batch size 500. Model: LLaMA3-8B.

### A.5. Hyper-Parameter Sensitivity Analysis

**Number of Candidate Numbers** In the main text, our empirical pipeline starts by sampling 500 candidates (`candidate number`) from the training prompts, and then randomly generates 20000 pairs of comparisons using either in-prompt comparison or cross-prompt comparison. Then, we select `annotation batch size` number of comparisons to annotate. In this section, we evaluate the performance difference by using a larger `candidate number` 1000.

In experiments, we find those setups do not significantly change the performance of different methods. The performance of D-opt and Past-Aware D-opt are especially robust to those hyper-parameter choices.

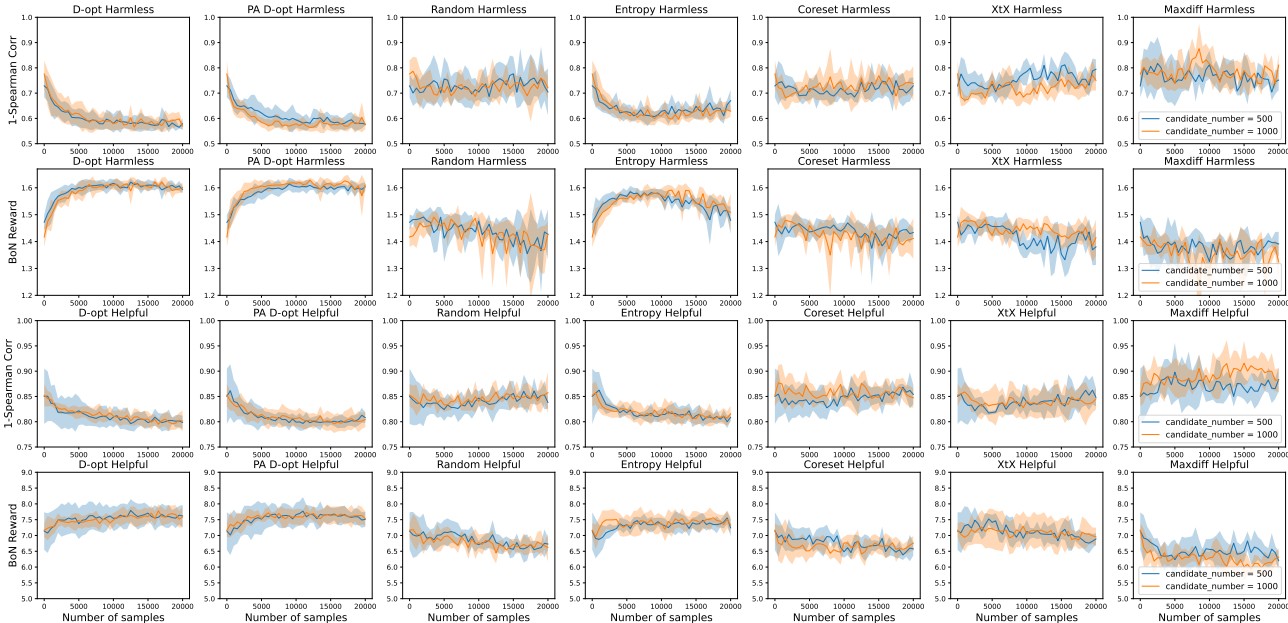

Figure 19: Preference annotation with different `candidate number` choices. Annotation batch size 500. Model: Gemma2B.

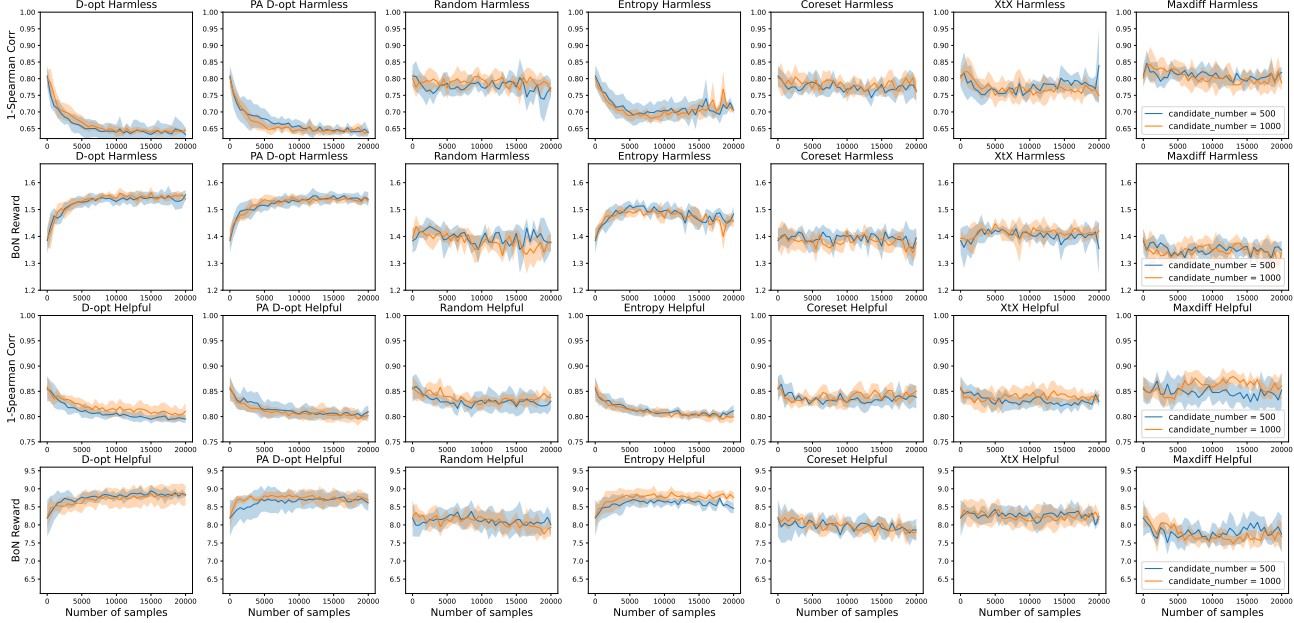

Figure 20: Preference annotation with different `candidate number` choices. Annotation batch size 500. Model: Gemma7B.

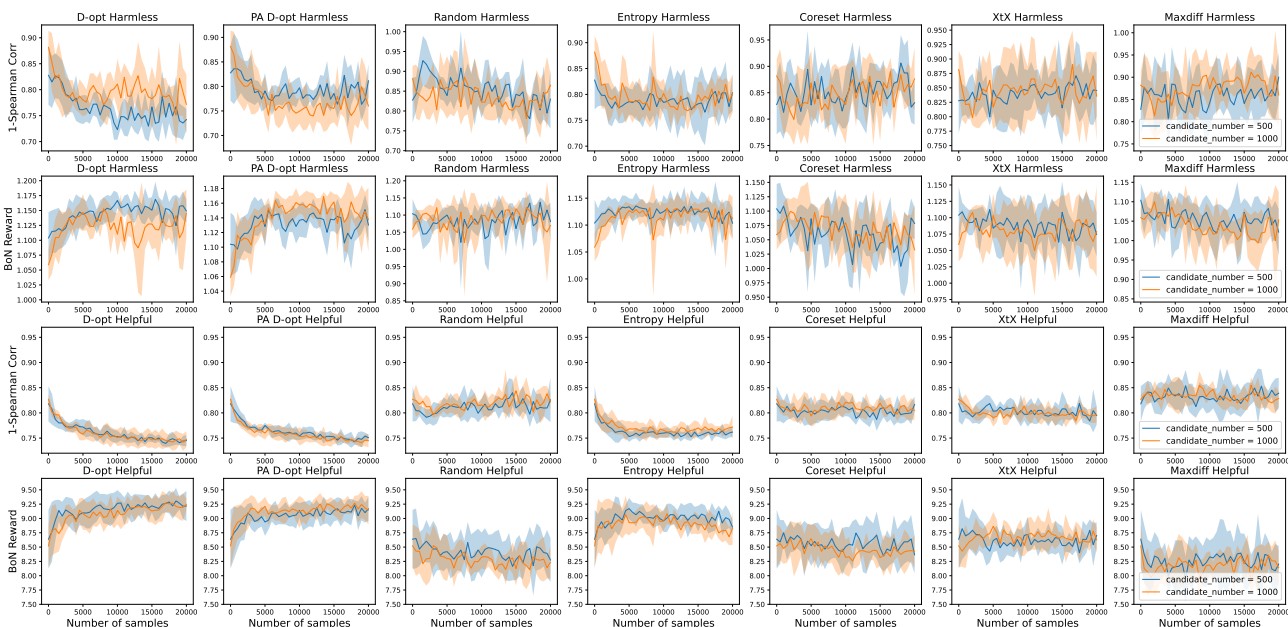

Figure 21: Preference annotation with different `candidate number` choices. Annotation batch size 500. Model: LLaMA3-8B.

**Number of Hidden Units in 3-Layer MLPs**    In all main text experiments, we use 3-layer MLPs with 64 **hidden units**. In this section, we evaluate the performance difference by using a larger `hidden unit` 128.

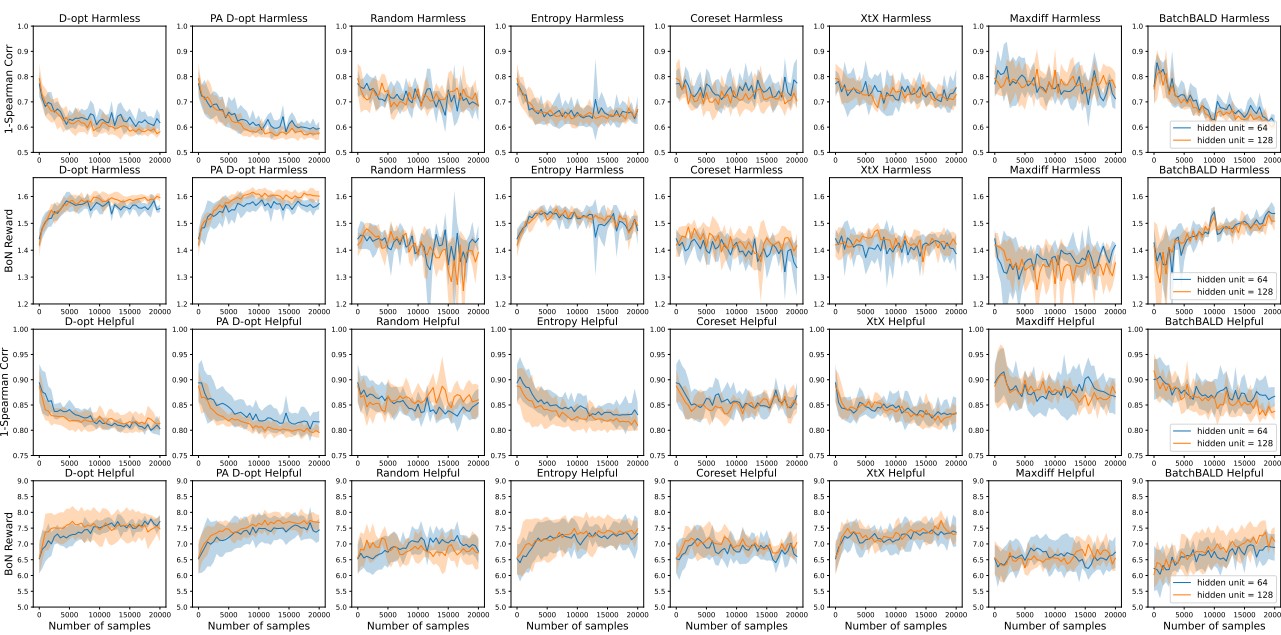

Figure 22: Experiments with different `hidden unit` choices. Annotation batch size 500. Model: Gemma 2B.

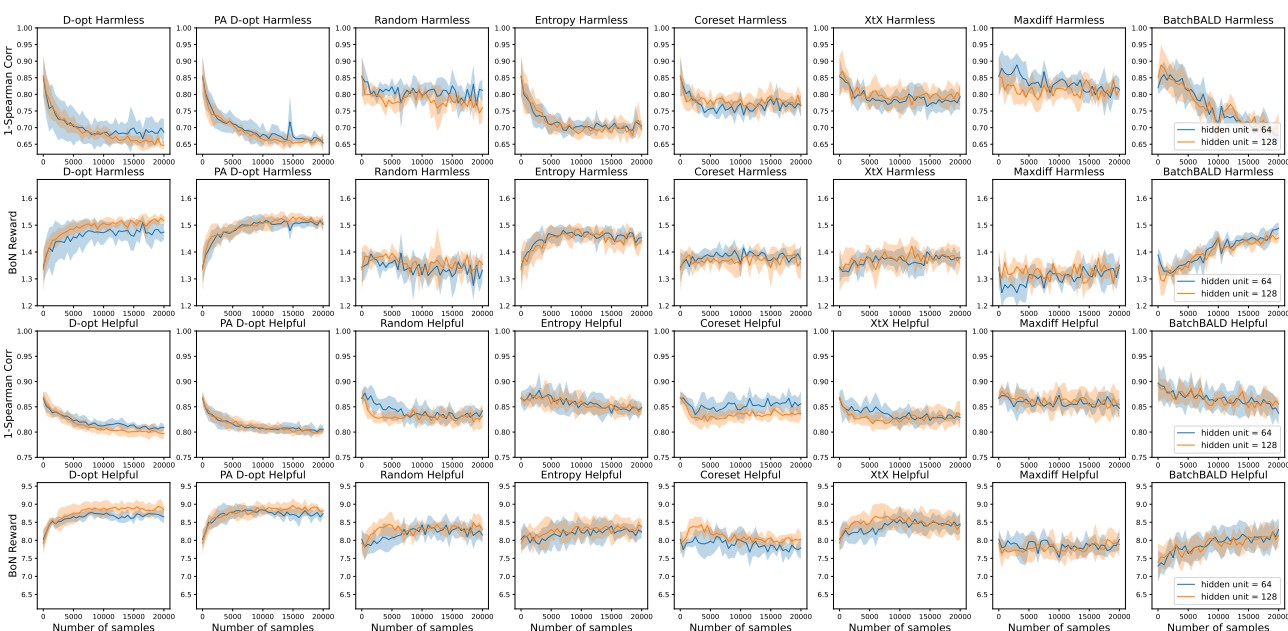

Figure 23: Experiments with different `hidden unit` choices. Annotation batch size 500. Model: Gemma 7B.

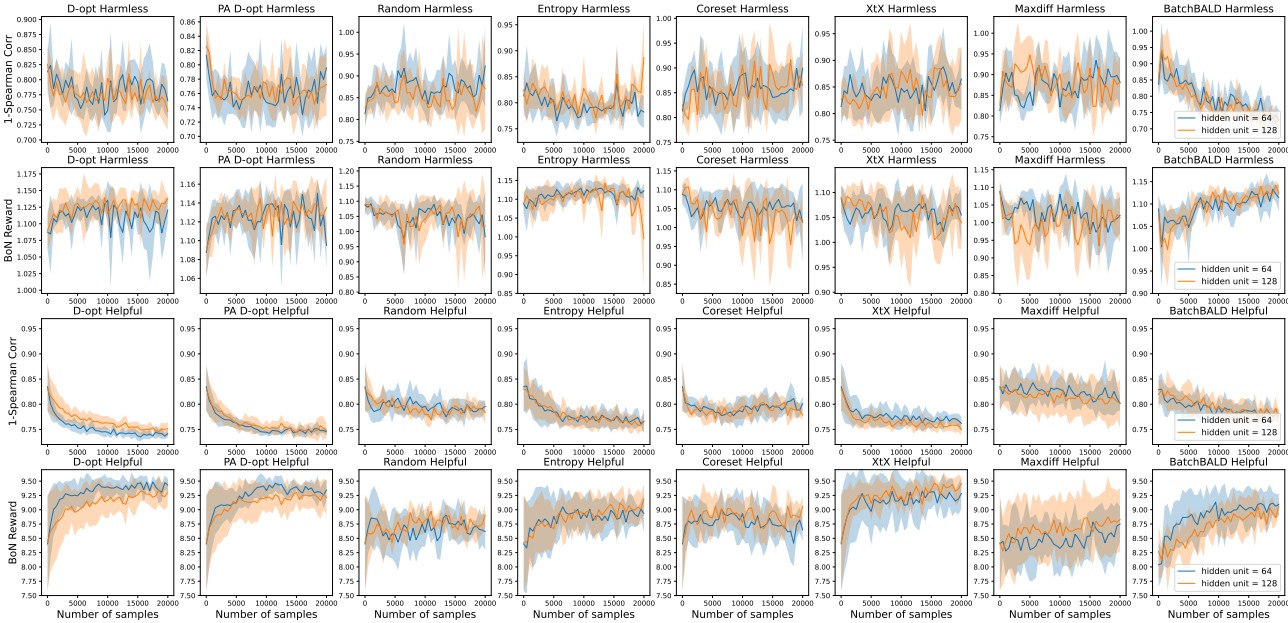

Figure 24: Experiments with different `hidden unit` choices. Annotation batch size 500. Model: LLaMA3-8B.

## B. Further Notes on Cross-Prompt Annotations.

Cross-Prompt annotations were explored as a way to increase annotation quality by Sun et al. (2025a) and empirically studied in Yin et al. (2024). A natural question is whether this is possible in practice. If one is willing to assume there exists a scalar value reward function, and human comparisons are based on that function, then cross-prompt is possible because each prompt-response pairs are assigned a real value that are comparable to each other. A single-word change in the prompt without changing its meaning likely will not change what responses are helpful or harmful and make these pairs, even if cross-prompt, comparable. It is possible, however, that the reward function is very rough in changing prompts, making the reward function for one prompt not transferable to the other and making it hard to get a better response for one prompt using a reward function learned from other prompts. Even though, if one is willing to believe that prompts live in some lower-dimensional manifold and the reward function acquires some regularity in that space, Cross-Prompt annotations might help better learn these dependencies.

