# OpenReview forum: "Active Reward Modeling: Adaptive Preference Labeling for Large Language Model Alignment"
_ICML.cc/2025/Conference — ICML 2025 poster_

### Official Review · Reviewer_Hkj8 · 2025-03-05

**Overall Recommendation:** 3

**Summary:**

This paper aims to enhance the reward model in RLHF. Drawing inspiration from active learning, the authors propose Fisher information-based selection strategies to construct an ideal comparison dataset.
The experiments show the effectiveness of the proposed method.

## update after rebuttal
The authors' response have addressed my concerns, and I decide to raise my score.

**Claims And Evidence:**

Yes

**Essential References Not Discussed:**

None.

**Experimental Designs Or Analyses:**

No significant issues.

**Methods And Evaluation Criteria:**

The proposed method is conceptually sound.

However, the authors claim that their approach effectively balances the exploration of the representation space and facilitates informative comparisons between pairs. To substantiate these claims, it is essential to incorporate empirical metrics that quantify these aspects. For example, diversity metrics can be employed to measure the extent of exploration.

**Other Comments Or Suggestions:**

None

**Other Strengths And Weaknesses:**

Other Strengths

1.	The authors conduct experiments using multiple LLMs of varying sizes, demonstrating the method's applicability across different model scales.

Other Weaknesses

1.	Ambiguity in Main Contributions.
The primary contribution of this paper remains ambiguous, as the proposed method appears to be a straightforward application of active learning to the reward model without any significant modifications or innovations.

2.	Limited Applicability to the Bradley-Terry Model.
The proposed method is exclusively compatible with the Bradley-Terry model, which may restrict its applicability to other types of reward models[1]. This limitation poses concerns regarding the method's flexibility and its potential integration with alternative reward modeling frameworks.

3.	Inappropriate Placement of Related Works in Section 3.1.
Section 3.1 predominantly serves as a detailed introduction to various related works, which detracts from its placement within the methodology section.

4.	Increased Risk of Over-Optimization and Reward Hacking.
The proposed method may inadvertently elevate the risk of over-optimization and reward hacking. By fine-tuning the reward model through active learning strategies, there is a potential for the model to excessively optimize for specific rewards, leading to behavior that exploits loopholes or unintended shortcuts rather than genuinely achieving desired outcomes.

**Questions For Authors:**

1.	What are the key hyperparameters used in this paper, and how were they chosen or tuned?

2.	How does the proposed method adapt to scenarios where multiple types of preference data exist?

3.	What is the accuracy of the initial reward model on the dataset, and how does it compare to the final model?

4.	Have you experimented with datasets other than Anthropic? If so, what were the results?

5.	What is the time consumption of the proposed method, and how does it scale with larger datasets or more complex models?

**Relation To Broader Scientific Literature:**

The primary contributions of this paper pertain directly to the development and enhancement of reward models within the framework of RLHF.

**Theoretical Claims:**

The paper does not contain proofs.

---

> ### Author Rebuttal · Authors · 2025-04-01
>
> **We thank our reviewer for their time and effort devoted to improving our paper. We have carefully considered each point of feedback and will provide our point-by-point responses below.**
>
> ---
> ### **P1. Main Contributions**
> We thank the reviewers for raising the question and reminding us to further highlight our contributions.
> We would like to note a meaningful machine learning contribution typically involves (1) **improved empirical results** on a relevant task compared to strong baselines and (2) **insights into why the method works**. Our paper proposing active reward modeling meets both:
> 1) Empirical success – We adopt a well-established approach to a novel setting, outperforming baselines including [1] (ICML’2024) and the SoTA active learning method from [2].
> 2) Insight – We highlight the trade-off between exploring representation space and comparing uncertain pairs.
>
> While active learning is well-studied, our focus is on **applying a classical method to prompt-response selection for reward modeling**. Classical approaches offer robustness, interpretability, and simplicity—our method leverages these strengths while achieving SoTA performance.
>
> ### **P2. Applicability beyond the BT Model**
> We acknowledge the reviewer’s concern that our method is designed around the BT model. The reference [1] in the reviewer’s original review was missing – if the reviewer can provide it, we’d be happy to discuss it further.
>
> While improving BT is worthwhile, it remains a widely used framework for modeling human preferences due to its simplicity. Given its role in RLHF workflows, improving sample efficiency within this framework is valuable and lays the foundation for further extensions.
>
> Our approach—leveraging last-layer features and optimizing Fisher information (FI) —is also compatible with other neural network-based statistical models, though evaluating broader applications is out of the scope of this paper.
>
> ### **P3. Related Works in Sec 3.1**
> We thank the reviewer and have moved them to the related work section in our revision.
>
> ### **P4. Over-Optimization and Reward Hacking**
> We understand the reviewer’s concern to be that active learning may lead to overfitting. Indeed, poorly designed active learning can cause overfitting— e.g., always selecting the least uncertain samples may degrade performance, sometimes performing worse than random sampling.
>
> Our FI-based approach mitigates this by encouraging **exploration in the embedding space**, ensuring selected samples remain diverse, and reducing the risk of overfitting to a narrow subset. Additionally, using last layer features—rather than all layers (as in [2])—acts as regularization, preventing over-optimization and improving robustness esp. in the early stage.
>
> ### **P5. Hyperparameters**
> The proposed method is not sensitive to its hyper-parameter choices, and we have extensive ablations studies on all hyper-parameters:
> - The reward model MLP architecture: ablation study in Appdx.A.4.
> - Batch size: ablation study in Appdx.A.2
> - Different base models: we test three of them in Sec 5.1
>
> ### **P6. Preference data types**
> The method is general --- the method can be applied as long as the FI can be calculated.
>
> ### **P7. Datasets**
> Resource constraints limited our experiments to the datasets we reported. While training reward models are cheap, benchmarking across 5 seeds x 32 hyperparameter settings x 3 LLMs x 8 methods led to a **3840× cost increase**, exceeding 2000 USD per dataset on cloud platforms. Despite this, our experiments provide strong empirical support for the method and we can draw statistically significant conclusions from those extensive empirical results.
>
> That being said, should there be important insights we need to draw from new experiments, we are more than happy to add them.
>
> ### **P8. Initial RM Performance**
> - The initial performance can be seen in e.g. Fig.3 at iteration 125 (for an initial model trained with 125 random samples). Using our strategy can lead to 5-20x higher sample efficiency as compared to the other baselines and more than 2x higher asymptotic performance.
>
> ### **P9. Time consumption**
> - The optimization problem for selecting data based on the linear approximation of FI requires a single forward pass to obtain the last-layer embedding and a backward pass to compute the FI, which is fast. Working on the embedding space, the experiment shown in Fig.3 took less than 10 minutes to finish for our method and baselines other than [2] and 2 hours for [2] on a CPU machine.
>
> ---
> **We thank the reviewer again for their effort in improving our work. If there should be any remaining concerns or questions, we are keen to do our utmost to address them. Please kindly consider increasing their rating if we address their concerns.**
>
> _**References**_
>
> [1] Active preference learning for large language models
>
> [2] Batchbald: Efficient and diverse batch acquisition for deep Bayesian active learning

---

> > ### Comment · Reviewer_Hkj8 · 2025-04-03
> >
> > Thank you for your response. I apologize for the missing references in the previous comments.
> >
> > I am curious about whether this method can be adapted to other types of reward models beyond BT models such as [1]. Can you provide some discussion?
> >
> > [1] Quantile Regression for Distributional Reward Models in RLHF. https://arxiv.org/abs/2409.10164

---

> > > ### Author Response · Authors · 2025-04-03
> > >
> > > We thank the reviewer for their clarification! [1] is indeed a promising model for learning human reward beyond point estimates. Yes, our strategy can be adopted with some modifications — the strategy can be seen as minimizing the _determinant of asymptotic covariance of the last layer weights_. There are general methods to calculate the covariance for M-estimators where the parameters are estimated by minimizing a loss function with some regularity. The quantile regression part in [1] falls in this general category [2]. The target is a similarly weighted version of the embedding difference. __Following the notation of [1], since all data points are equally weighted in eq (2) of [1] and assuming a centered residual the target asymptotic precision (inverse of the covariance) is proportional to $XX^\top/\tau(1-\tau)$, and we can use its determinant.__ The full general form of this covariance matrix and derivation is a bit involving and can be found in Chapter 3 of [2] using influence functions. For a fixed $\tau$ this is quite similar to det(XtX) we tested but the data and workflow for the reward model would be quite different so a different test is needed to tell.
> > >
> > > We have added discussions in our revision, and we believe it provides further insight and demonstrates the generality of our method.
> > >
> > > ---
> > > Please let us know if further clarification is needed.
> > > Should there be any remaining questions or suggestions, we are more than happy to further address them!
> > >
> > > ----
> > > _**References**_
> > >
> > > [1] Quantile Regression for Distributional Reward Models in RLHF. https://arxiv.org/abs/2409.10164
> > >
> > > [2] Koenker, R., 2005. Quantile regression (Vol. 38). Cambridge University Press.

---

### Official Review · Reviewer_M1YG · 2025-03-11

**Overall Recommendation:** 1

**Summary:**

The paper proposes active learning methods for reward modeling. These active learning methods work as follows:

1. For a large set of prompts, use LMs to generate responses for comparison.
2. Form these generated prompt-response pairs into tuples for either in-prompt comparisons (prompt, response 1, response 2) or cross-prompt comparisons (prompt 1, response 1, prompt 2, response 2).
3. Select a subset of tuples to annotate, using the current reward model.
4. Use the selected subset of tuples to update the reward model.
5. Return to step 1 and iterate.

The key problem is step 3: how to select the subset of comparisons from a large candidate pool. For this selection, the authors consider a few choices of scoring functions (Section 3.1), which assign a score to each subset of comparison tuples. Empirically, the scoring rule based on D-optimality works best, according to the authors' evaluation.

**Claims And Evidence:**

The authors' main argument for active learning of reward models is that it reduces training costs. However, I find the evidence lacking.

The reason I think so is that the current study focuses on reducing the cost of annotating examples, but at the expense of introducing significantly higher costs in the language model (LM) response sampling phase (Figure 2). Indeed, to reduce the overall training cost, a trade-off between **the cost of generating responses** and **the cost of annotating them** must be considered. While active learning methods may reduce the number of training examples (and thereby the annotations of these examples), it introduce the extra cost of sampling, too. Specifically, the active learning algorithm (Figure 2) requires sampling **a large amount of responses** from language models before discarding most of them and narrowing down to a smaller subset to train the reward model. Furthermore, this sampling occurs over many iterations, making this process even more expensive.

The repetitive sampling process, as well as the repetitive reward model training process, is unique to the proposed active learning approach and not present in standard training. It seems that the authors did not account for this part of the training cost. What exacerbates the cost situation for the proposed sampling-intensive active learning is that increasing the LM size could lead to higher sampling costs, even though the cost of preference annotation by human experts remains constant. For this reason, I believe that the cost advantage of the authors' sampling-based active learning approach over standard reward modeling will at best narrow---and more likely reverse---as we scale up the model size to a point where the over-sampling of responses becomes costly enough to negate the benefit of annotating fewer responses.

Based on my reasoning above, I believe a comprehensive cost-performance trade-off comparison between various active learning methods and a standard reward modeling baseline (without active learning or sampling) is needed to validate the argument that active learning reduces training costs. Currently, such cost-performance trade-off experiment is lacking.

**Essential References Not Discussed:**

Not applicable.

**Experimental Designs Or Analyses:**

See "Claims And Evidence".

**Methods And Evaluation Criteria:**

See "Claims And Evidence".

**Other Comments Or Suggestions:**

A few typos in the paper:

line 56: $r \in \mathbb{R}^D \to \mathbb{R}$ should be $r: \mathbb{R}^D \to \mathbb{R}$.

line 4 of Algorithm 1: If I am not mistaken, the argmax should be over $C \subset \mathcal{P}\_{s}$ instead of  $C \subset \mathcal{P}\_{s-1}$.

**Other Strengths And Weaknesses:**

The main weakness, as detailed in "Claims And Evidence", is that it is unclear how much active learning can lower the overall training cost of reward model. Cost--performance tradeoff experiments are needed for the argument of reducing the training cost to be convincing.

**Questions For Authors:**

I am wondering whether the following approaches are valid for the cost-performance tradeoff comparison:

1. Assign a constant value to represent the cost for each preference annotation.
2. Measure the FLOPS required for training the active learning procedure (Figure 2), which should include both the reward model training cost and LLM sampling cost.
3. Compare the overall cost (annotation cost and training FLOPS) of active learning with that of standard reward model training methods.

**Relation To Broader Scientific Literature:**

This paper aims to train a reward model using active learning. As the reward model is a binary classifier, the proposed approach is not specifically tailored to natural language processing. The literature on active learning for (binary) classification is most relevant, which is very broad.

**Theoretical Claims:**

Not applicable, as the paper does not contain theoretical claims or proofs.

---

> ### Author Rebuttal · Authors · 2025-04-01
>
> **We thank the reviewer for investing their time in reviewing our work, and providing insightful suggestions for improving our paper. We have carefully considered each point of feedback and provided our point-by-point responses below.**
>
> ---
> ### **P1. Response to the main concern: cost of oversampling**
> _We thank the reviewer for raising the question on sampling cost, and we are reminded there could be misunderstandings about the generation cost in the proposed procedure._
>
> **1. Sampling Cost**
>
> While our algorithm requires a **large number of comparisons** to choose from, these comparisons are generated **combinatorially from a small number of responses**, this is because we can reuse a single response in multiple comparisons.
> For instance, generating $10$ responses rather than $2$ responses as in non-active random sampling will lead to $ 45 $ times more comparisons. Moreover, in the cross-prompt comparisons setup, $10$ responses per prompt on $500$ prompts lead to $12M$ potential comparisons.
>   - [**Spend 2.5 more USD in generation**] To compare the cost of generation and annotation. With our setup of 500 prompts, using the DeepSeek API at off-peak hours, generating 2 responses would cost 0.7 USD (2M input tokens and 2M output tokens); and generating 10 responses would cost 2.5 more USD.
>   - [**Saving 1000 USD with efficient annotation**] In our experiments, we find using 10 responses per prompt can significantly improve the annotation efficiency (by over 20x more efficiently as compared to only generating 2 responses). Annotating $20000$ randomly selected preference data will cost 160 labor hours (assuming 125 annotations per hour) and will cost more than 1000 USD according to the 7.5 USD minimum wage in the US. With our method, the same reward modeling performance can be achieved with $1000$ annotations, which cost $60$ USD (cf. Fig.3).
>
>     We will be more clear in the revision on this aspect.
>
> **2. Training Cost**
>
> As for the cost of re-training, we would note the fact that training reward models can be much more computationally efficient than fine-tuning language models [1-4]. In our work, we follow the embedding-based reward modeling setups and the computational cost is negligible as compared to the cost of LLM generation (i.e., **cost of RM training << cost of generation << cost of annotation**)
>
> In terms of FLOPs, training a BT reward model with $2048$-dim input on $1000$ samples and optimizing for $10$ epochs correspond to $1e11$ FLOPs. Generating 2 responses (1000-token-long) on those $1000$ samples using a 7B LLM needs $1e13$ FLOPs. The expense of finishing those computations on prevailing cloud service platforms costs less than **$0.01$ USD**
> On the other hand, hiring annotators for annotation is much more expensive. Annotating preference over $1000$ samples will induce a much higher cost.
>
> - The number of candidate prompts and responses per prompt can be tuned to accommodate different budget constraints. With more candidates, it is more likely to get more informative comparisons (c.f. Fig. 18-19 in Appendix A.5)
> - We also want to highlight that in our experiments, Fisher Information-based active training can achieve better asymptomatic performance compared to other methods for training, especially the most widely adopted random sampling approach.
>
> - The reward models can be updated more frequently than the LLMs. This is why our work focuses on active reward modeling rather than a full active alignment workflow (i.e., including both reward model training and LLM fine-tuning). As we have analyzed above, the cost of training reward models is significantly lower than the cost of response generation and collecting preference annotations. On the other hand, from the practical perspective, active (and more frequently updated) reward models can quickly adapt to user preferences and improve user experiences (c.f., Fig.3).
>
>
> ### **P2. Typos**
> We thank the reviewer for spotting them. We have fixed them in our revision.
>
> ---
> **Once again, we thank our reviewer for their effort in improving our work. If there should be any remaining concerns or questions, we are keen to do our utmost to address them. Please kindly consider increasing the rating if their concerns are well addressed.**
>
>
> ---
> _**References**_
>
> [1] Gao, Leo, John Schulman, and Jacob Hilton. "Scaling laws for reward model overoptimization." International Conference on Machine Learning. PMLR, 2023.
>
> [2] Go, Dongyoung, et al. "Compositional preference models for aligning LMs." The Twelfth International Conference on Learning Representations. 2024.
>
> [3] Sun, Hao, Yunyi Shen, and Jean-Francois Ton. "Rethinking reward modeling in preference-based large language model alignment." The Thirteenth International Conference on Learning Representations. 2025.
>
> [4] Barreto, André, et al. "Capturing Individual Human Preferences with Reward Features." arXiv preprint arXiv:2503.17338 (2025).

---

> > ### Comment · Reviewer_M1YG · 2025-04-08
> >
> > Thank you for your response. The back-of-the-envelope computation is very helpful. However, could you clarify why the DeepSeek API was used as a metric for computing the sampling cost? Your submission didn't mention DeepSeek. My understanding is that your paper used Gemma2b, Gemma7b, and LLaMA3-8b as the models that generate responses. Is this understanding correct?
> >
> > >With our method, the same reward modeling performance can be achieved with 1000 annotations, which cost 60 USD (cf. Fig.3).
> >
> > Could you walk me through the computation here? Does 1,000 annotations mean 1,000 responses or 10 * 1000 responses? Suppose it is the former (1,000 responses), then I suppose the cost via the DeepSeek API is 1000 * (0.7 * 2.5) = 320 and not 60. Where is my mistake here?
> >
> > > As for the cost of re-training, we would note the fact that training reward models can be much more computationally efficient than fine-tuning language models.
> >
> > Upon re-reading the paper, I understand that the authors use MLP-based reward models with few (like 3) hidden layers throughout the evaluation. Is this understanding correct?
> >
> > If so, provided that the reward models are shallow MLPs, I agree the cost of re-training them is cheap in the authors' setting. However, in realistic post-training settings, reward models are usually language models with a classifier head and not MLPs; see, e.g., sequence classifier reward models in RewardBench (https://github.com/allenai/reward-bench). The repetitive training approach could cause additional computational overhead when applied to these larger models.
> >
> > Furthermore, is there a reason why the authors chose to use an MLP-based reward model? To me, using an MLP as a reward model is quite unconventional choice. As even the arguably the first RLHF paper [Stiennon et al, (2020)](https://arxiv.org/abs/2009.01325) uses LM-based reward models. In practice, it seems unlikely that one would be willing to sample thousands of responses from language model APIs for active learning, yet be unwilling to use an LM-based reward model to improve accuracy. In the paper, it was briefly mentioned that "To separate representation learning from reward modeling, we train our reward model using joint embeddings of prompts and responses." But as long as the reward model is based on the same checkpoint of LM, "representation learning" is not really a confounder that influences the reward modeling accuracy, right? Plus, this is the more realistic setting that practitioners use (see, e.g., the RewardBench paper I mentioned earlier).

---

### Official Review · Reviewer_kGRg · 2025-03-11

**Overall Recommendation:** 4

**Summary:**

The paper presents an evaluation of different methods to determine which samples in a dataset of (prompt, response one, response two) triplets should receive a preference label and be used to train a LLM reward model using the Bradley-Terry model. The authors propose to use a modification of D-optimality on the embedding space of the reward model to determine which samples to label and train with. Six other strategies to select samples are compared against with a detailed description of each method to select the training samples. The difference between the methods is examined with a simple, 2-dimensional toy dataset and a the Helpful and Harmless data from Anthropic. For the Helpful and Harmless 3 different LLMs are compared to understand how the LLM impacts the data selected with each strategy. The experiments and results rely on 1 - Spearman correlation and Best of N reward. Performance is evaluated both within and across prompts, and an experiment is run to look at the how the number of samples impacts reward model performance. The learned reward models are compared to Gold Standard reward models for the HH Anthropic dataset. The experiments demonstrate the benefits of D-Optimality over the other methods.

**Claims And Evidence:**

The claims made in the submission are supported by clear and convincing evidence. However, the paper would be stronger if at least one other dataset (e.g. OpenAssistant) was evaluated as well.

**Essential References Not Discussed:**

The references discussed appears sufficient

**Experimental Designs Or Analyses:**

Yes, I checked the soundness/validity of the experimental design and analysis. I checked the set up of the 2-dimensional toy experiment and of the helpfulness and harmless experiments. Although many important details are provided to understand the experimental set up, there are some gaps. Please see the questions below. The impact of the number of samples is assessed as well as the impact of annotation size for the D-optimality sample selection strategies.

As there is evidence that reward modeling abilities does not always directly translate to final policy performance, an evaluation of the quality of policy the reward model is able to train is missing. This can be done either with DPO or PPO.

It would be great to supplement the comparison of samples selected by the different sample selection strategies for the 2-dimensional toy domain experiment with an evaluation of the quality of the learned distribution to draw the connection between the samples selected and what the reward model learns.

**Methods And Evaluation Criteria:**

Yes, the proposed methods and evaluation criteria make sense for the problem at hand.

**Other Comments Or Suggestions:**

1. Please more clearly highlight that D-Optimality is the method you are "proposing" in this paper. It takes a bit of re-reading to understand that this part is a main contribution.
2. It is difficult to interpret and draw conclusions from Figure 1. Including a table in the appendix with numbers detailing the attributes mentioned in the discussion would be helpful. For example, the mean and standard deviation in number of connections between each point, the difference in rewards, and a measure of sample diversity.

**Other Strengths And Weaknesses:**

The paper is well written and easy to follow with key take aways clearly stated along with key aspects of evaluation, such as success criteria clearly state.

**Questions For Authors:**

1. What is I^s in the active learning section where the datasets are defined?
2. What is meant by "joint embeddings of prompts and responses" on line 326? Please expand upon this.
3. For figures 3, 4, and 5, what are the error bars over?

**Relation To Broader Scientific Literature:**

The paper is well situated within the broader scientific literature on preference sample selection.

**Theoretical Claims:**

There are not proofs to assess

---

> ### Author Rebuttal · Authors · 2025-04-01
>
> **We thank our reviewer for their encouraging feedback. To respond to the point raised by this reviewer, below, please find our answers to each of the questions.**
>
> ---
>
> ### **Q1. Translating performance of reward model not to alignment**
>
> - In our experiments, we evaluated the effectiveness of different reward models using Best-of-N (BoN) sampling following [1,2] and the Spearman Ranking correlation following [3]. This choice was driven by three key considerations:
>     1. **Performance**: Empirical studies show BoN achieves better performance than PPO [1,2,4,5]
>     2. **Stability and Reduced Engineering Overhead**: BoN requires no hyperparameter
> tuning and is more stable than PPO, leading to more consistent and interpretable results. [1,6,7]
>     3. **Computational Efficiency and Reproducibility**: BoN’s reusability across N generations during test time makes it more computationally efficient compared to policy gradient optimizations. In contrast, using PPO or DPO for our experimental setups (in total 3840 = 5 random seeds x 3 LLMs x 8 methods x 2 annotation strategies x 4 batch sizes x 2 network sizes x 2 candidate sizes) would be computationally prohibitive since each setup requires distinct LLM fine-tuning [8].
>
> - It is indeed true that a good reward model is not sufficient for good performance. However, it is probably necessary for a good final policy performance and the best-of-n sampling we tested can be seen as a best-case scenario [1] that can be treated as an approximated upper bound of final performance after RL that does not involve costly DPO or PPO.
>
>
> ### **Q2. Adding toy example performance figure**
> We thank our reviewer for the great idea! We do have a similar performance figure and we will add it in the revision of the paper.
>
>
> ### **Q3. More clearly highlight that D-Optimality**
> We thank our reviewer for their suggestion. We agree highlighting more on the D-Optimality in the introduction, and method in our revision can further enhance its clarity.
>
> ### **Q4. Conclusions from Figure 1**
> We thank our reviewer for the great suggestion. We will add a table detailing the number of connections and sample diversity, e.g., the variance of pairs in the original space and the last layer of the neural net
>
> ### **Q5. What is $I^s$ in the active learning section where the datasets are defined?**
> The number of unannotated prompt-response pairs. We have clarified the notations accordingly in our revision.
>
> ### **Q6. expand upon "joint embeddings of prompts and responses" on line 326**
> By joint embeddings, we were referring to the embeddings of prompt-response combinations.
>
> ### **Q7. For figures 3, 4, and 5, what are the error bars over?**
> In our experiments, all error bars are generated with **5 repeated runs with different random seeds** to show the statistical significance of performance differences.
>
> ---
> **We thank the reviewer again for their effort in improving our work. If there should be any remaining concerns or questions, we are keen to do our utmost to address them.**
>
> ---
>
> _**References**_
>
> [1] Gui, Lin, Cristina Gârbacea, and Victor Veitch. "Bonbon alignment for large language models and the sweetness of best-of-n sampling." arXiv preprint arXiv:2406.00832 (2024).
>
> [2] Gao, Leo, John Schulman, and Jacob Hilton. "Scaling laws for reward model overoptimization." International Conference on Machine Learning. PMLR, 2023.
>
> [3] Sun, Hao, Yunyi Shen, and Jean-Francois Ton. "Rethinking reward modeling in preference-based large language model alignment." The Thirteenth International Conference on Learning Representations. 2025.
>
> [4] Dong, Hanze, et al. "Raft: Reward ranked finetuning for generative foundation model alignment." arXiv preprint arXiv:2304.06767 (2023).
>
> [5] Yuan, Zheng, et al. "Rrhf: Rank responses to align language models with human feedback without tears." arXiv preprint arXiv:2304.05302 (2023).
>
> [6] Ivison, Hamish, et al. "Unpacking dpo and ppo: Disentangling best practices for learning from preference feedback." Advances in neural information processing systems 37 (2024): 36602-36633.
>
> [7] Xu, Shusheng, et al. "Is dpo superior to ppo for llm alignment? a comprehensive study." arXiv preprint arXiv:2404.10719 (2024).
>
> [8] Stiennon, Nisan, et al. "Learning to summarize with human feedback." Advances in neural information processing systems 33 (2020): 3008-3021.

---

> > ### Comment · Reviewer_kGRg · 2025-04-07
> >
> > Thank you for your responses. I will be leaving recommendation as accept.

---

### Official Review · Reviewer_6Pdx · 2025-03-14

**Overall Recommendation:** 2

**Summary:**

This paper investigates strategies to leverage adaptive preference labeling for reward modeling in LLM alignment. The authors propose an Active Reward Modeling ARM framework that uses Fisher information to score and select informative preference comparisons to improve annotation efficiency; then, they benchmark several active learning and experimental design-based strategies across multiple models and datasets. They report gains in annotation efficiency and reward model performance compared to random sampling and other baseline methods.

**Claims And Evidence:**

The authors' claims regarding the efficiency of D-optimal and past-aware D-optimal methods are supported by experimental results, but the paper lacks theoretical analysis to explain why these methods outperform others in reward modeling and how they impact the process.

**Essential References Not Discussed:**

The main limitation of this paper is the lack of discussion on the motivation and theoretical justification of the proposed method, rather than missing references.

**Experimental Designs Or Analyses:**

The experimental design relies on Fisher information-based selection during the preference data sampling process, with the assumption that it consistently leads to better reward model performance. The paper does not address the potential biases introduced by focusing on "informative" but possibly unrepresentative comparisons. It would be valuable to discuss the trade-off between focusing on hard-to-predict samples and maintaining representativeness of human preferences, and the actual improvements in real-world alignment this strategy offers. Specifically, how does selecting for Fisher information impact the generalizability and alignment quality of the reward model?
The comparison of ARM's preference data selection and resulting reward model performance to baselines is not sufficiently explored. The experimental analyses focus on annotation efficiency and ranking accuracy but do not assess downstream alignment impacts or broader generalization. These are limited and do not offer enough evidence to assess ARM's real-world alignment capabilities. Given that reward models are often applied to diverse and complex downstream alignment tasks, and that preference data may not always map directly to better alignment, it would be beneficial to provide a broader set of downstream evaluations or human-in-the-loop assessments. This would offer a more comprehensive view of ARM's effectiveness in practical LLM alignment scenarios.

**Methods And Evaluation Criteria:**

Generally appropriate.

**Other Comments Or Suggestions:**

The authors do not clearly explain why Fisher information is the most suitable criterion for preference data selection in the context of reward modeling, compared to other uncertainty or diversity-based approaches.

I wonder why selecting "informative" comparisons based on Fisher information would align better with real human preferences, especially when human preference data may be noisy or inconsistent. Could the authors clarify the motivation behind this assumption?

**Other Strengths And Weaknesses:**

The idea of actively selecting informative preference comparisons for reward modeling is novel. Most existing work focuses on random sampling or heuristic-based selection for preference data, but these approaches cannot efficiently optimize the annotation process to maximize learning efficiency, this could affect the scalability and effectiveness of reward models in LLM alignment. This paper introduces a method that leverages Fisher information-based selection strategies to identify the most informative preference comparisons, as well as significantly improves annotation efficiency and reward model performance with fewer labeled comparisons.

**Questions For Authors:**

See above.

**Relation To Broader Scientific Literature:**

This framework builds on prior work in active learning, experimental design, and preference modeling, specifically leveraging ideas from Fisher information-based experimental design and Bradley-Terry preference models to improve reward model data efficiency. And this paper compares its Active Reward Modeling framework against several baselines, including random sampling, max reward difference, max entropy, batchBALD, and coreset methods for preference data selection.

**Theoretical Claims:**

For assumptions, I notice two potential issues:

i. The paper assumes that Fisher information computed over the last layer of the reward model sufficiently captures the informativeness of preference comparisons. But this may be an overly idealized assumption. Actually, deep reward models exhibit complex, non-linear behaviors. If we focus only on the last layer, we may overlook critical interactions and uncertainties in earlier layers, and this could limit the robustness of the active selection strategy.

ii. The paper assumes that reward model improvement is directly aligned with selecting samples that maximize Fisher information. However, this does not guarantee that the selected comparisons will improve generalization to downstream tasks. In particular, maximizing Fisher information might bias the reward model toward "hard" comparisons that are not representative of typical user preferences. I worry that it may lead to a misaligned reward signal.

---

> ### Author Rebuttal · Authors · 2025-04-01
>
> **We thank our reviewer for their careful and detailed review of our work. We appreciate that several valuable points of feedback were included, and we believe that the updated version of our work will be strengthened by reflecting on these points. We address concerns and questions below.**
>
> ---
> ### **Q1. Can using the last layer overlook complexity and lost robustness**
> We thank our reviewer for raising the insightful question. An extension using uncertainty of more parameters via Fisher Information (FI) is possible. E.g. [2] used Bayesian uncertainty for all parameters. In our experiments we find our method to be better than [2]. An intermediate approach using uncertainty of more layers than the last could potentially further improve the performance yet is left for future research.
>
> ### **Q2. Can actively selected samples lead to better generalization**
> We agree with our reviewer that evaluating the generalization ability of RMs is necessary. In our work, we evaluated our RMs with holdout data, showing no signs of overfitting (in contrast, the entropy sampling method shows overoptimization). Relating to the previous point, using only the last layer features may be a regularization, preventing over-optimization. This could explain why we outperform [2].
>
> ### **Q3. Downstream performance**
> - We agree that a good reward model may not lead to improved alignment. Human preferences are complex, and no scalar reward function can fully capture it. While this is a valid concern, addressing it is beyond the scope of this paper. We follow the common RLHF assumption that a reward function serves as a sufficiently useful optimization objective. Ideally, we would test the entire alignment pipeline but such a test is beyond our computational resources. We believe our experimental setup provides sufficient evidence of the method's utility.
> - To be more concrete, our evaluation setup has
>    - **Operational definition of alignment** We define alignment as having high human reward value. Since the true human reward is not directly accessible, we use a "golden" reward model—trained on a large dataset—as a surrogate.
>    - **Metric of success** An active learning method is successful if best-of-N samples based on this reward model achieve higher rewards with fewer human annotations. We also reported the Spearman correlation of RMs and the golden reward models.
>
>
> ### **Q4. FI Suitability**
> - Intuitively, this approach balances exploration in representation space, enabling the reward model to predict across a broader range of prompt-response pairs while exploiting uncertain comparisons between hard-to-distinguish pairs. This is preferable to purely diversity-based methods, which may focus on “easy” comparisons, and more effective than purely uncertainty-based methods, which often compare similar pairs, limiting generalization (c.f. Fig.1).
> - Empirically, our approach outperforms pure diversity methods, e.g. coreset, and pure uncertainty method e.g. entropy sampling [1]. Our method achieves both better sample efficiency and asymptotic performance.
>
> ### **Q5. Does FI Align Better with Real Human Preferences?**
> - This is a valid concern, and we break it down into two parts as FI depends on model:
>    1) Is Bradley-Terry (BT) a good model for human preference?
>    2) If we accept BT as a good model can FI-based active learning handle noise and improve sample efficiency?
> - For 1), this is a valid concern. As George Box said, "all models are wrong, but some are useful." BT is: a) widely adopted, and b) proven to be effective in large-scale applications. Thus, we consider it a useful model to improve sample efficiency. Testing alternative models to account for inconsistencies e.g. transitivity violations is beyond the scope of this paper.
> - For 2), assuming BT is useful, our experiments demonstrate that FI-based sampling design can yield a better-performing reward model with less data. The intuition behind is discussed in the response to why FI is a good metric.
>
> To summarize: since BT is widely adopted and acquiring human preference data is expensive, our goal is to improve sample efficiency by carefully selecting comparisons.
>
> ---
> **We thank the reviewer again for their effort in improving our work. If there should be any remaining concerns or questions, we are keen to do our utmost to address them. Please kindly consider increasing the rating if their concerns are properly addressed.**
>
> ---
> _**References**_
>
> [1] Muldrew et al. "Active preference learning for large language models." ICML’24
>
> [2] Kirsch et al. Batchbald: Efficient and diverse batch acquisition for deep Bayesian active learning. NeurIPS’19

---

### Decision · Program_Chairs · 2025-05-01

**Decision:**

Accept (poster)

**Comment:**

This paper applies ideas in active learning to improving data selection for training reward models for LLMs. Reviewers were broadly supportive of acceptance. The main objection is from reviewer M1YG, who is focused on the questions of whether the cost of getting labels is really the relevant problem. I agree this is an important consideration. However, there are certainly cases where the labels are the dominant cost (e.g., very difficult problems or applications in protein LLMs), so I find the active learning problem to be well motivated.